# Mnemonic prediction errors bias hippocampal states

Oded Bein [1✉], Katherine Duncan[2] & Lila Davachi [3,4✉]

When our experience violates our predictions, it is adaptive to upregulate encoding of novel information, while down-weighting retrieval of erroneous memory predictions to promote an updated representation of the world. We asked whether mnemonic prediction errors promote hippocampal encoding versus retrieval states, as marked by distinct network connectivity between hippocampal subfields. During fMRI scanning, participants were cued to internally retrieve well-learned complex room-images and were then presented with either an identical or a modified image (0-4 changes). In the left hemisphere, we find that CA1-entorhinal connectivity increases, and CA1-CA3 connectivity decreases, with the number of changes. Further, in the left CA1, the similarity between activity patterns during cued-retrieval of the learned room and during the image is lower when the image includes changes, consistent with a prediction error signal in CA1. Our findings provide a mechanism by which mnemonic prediction errors may drive memory updating—by biasing hippocampal states.

[1] Department of Psychology, New York University, New York, NY 10003, USA. [2] Department of Psychology, University of Toronto, Toronto, ON M5S 3G3, Canada. [3] Department of Psychology, Columbia University, New York, NY 10027, USA. [4] Center for Biomedical Imaging and Neuromodulation, The Nathan S. Kline Institute for Psychiatric Research, Orangeburg, NY 10962, USA. ✉email: oded.bein@nyu.edu; ld24@columbia.edu

As our day unfolds, much of what we encounter is expected: we typically navigate to work or school along the same route, sit in the same seats in the same space and engage with the same people. However, layered on top of the repetition of similar places and events are novel or surprising events; and when we travel to unfamiliar places, we experience even more novelty. This interplay between similarity and novelty poses different demands on our memory system. On the one hand, the repeating aspects of each day may trigger the retrieval of related memories that may allow those memories to then serve as predictions to guide adaptive behavior[1–3]. By contrast, surprising events may shift the memory system toward encoding of those contextually novel events[4–8]. Intriguingly, the hippocampus has been proposed to mediate both the encoding of new events and the retrieval of previous related experiences[9–12]. However, at a mechanistic level, these processes require seemingly conflicting processes: new encoding benefits from plasticity in hippocampal networks while this kind of plasticity during retrieval may permanently alter the veracity of long-term memories[5,13,14]. Furthermore, at the neural population level, encoding presumably requires that current experiences be represented in an activity pattern distinct from other stored memories, a process known as pattern separation[14,15]. Retrieval, on the other hand, may be supported by the recovery of a previously encoded activity pattern, or pattern completion[10,14,16,17]. Thus, a critical question is how can the hippocampal system balance these two seemingly opposing processes? And what factors may bias the hippocampus towards one over the other[4–6,18–20]?

Current models of hippocampal function propose that communication along distinct CA1 pathways may be associated with encoding and retrieval states[5,13,18]. Specifically, it has been proposed that, during encoding of novel experiences, input from the medial temporal cortical regions that receive numerous sensory inputs, such as the entorhinal cortex[21–24], may be prioritized by hippocampal area CA1. By contrast, during retrieval, CA1 may preferentially process input from hippocampal area CA3. CA3 neurons are highly interconnected, a feature proposed to facilitate pattern completion and promote the retrieval of encoding-related ensembles, which can then be conveyed to area CA1[10,14,17,25–28]. Empirical work in rodents has shown that CA1–entorhinal coherence is higher in the fast-gamma band compared with the slow-gamma band, while CA1–CA3 coherence is higher in the slow versus fast-gamma band, supporting a functional distinction between these pathways[19,29]. These different gamma band frequencies have also been linked to different behaviors, such as fast or slow running speed[18,29,30]. More recently, CA1 fast-gamma band activity was observed during learning of spatial routes in a maze, compared with slower-gamma activity evident during retrieval of learned routes[31]. Similarly in humans, Griffiths et al.[32] recently have shown with intracranial electroencephalogram (EEG) in the hippocampus an increase in fast-gamma activity during associative encoding compared with retrieval, in contrast to a complementary increase in slow-gamma activity during retrieval relative to encoding. Further, CA1–CA3 coherence has been shown to be enhanced in the central arm of a T-maze, potentially reflecting retrieval of the goal location[25]. Additional support of the dissociation between the two pathways comes from studies showing that CA1 coupling with the entorhinal cortex and area CA3 occurs at different phases of a theta cycle[5,13,33–35]. Extending this theoretical and empirical framework to humans, we have recently shown, using functional magnetic resonance imaging (fMRI), that CA1–CA3 functional connectivity is significantly enhanced during episodic memory retrieval compared with novel associative encoding[20]. Importantly, the magnitude of CA1–CA3 connectivity during retrieval predicted retrieval success[20]. Together, these results provide support for the idea that

the hippocampus may shift between encoding and retrieval states by modulating CA1 connectivity with distinct input regions.

One prominent factor that may bias hippocampal dynamics toward encoding rather than retrieval is mnemonic prediction error[6,8,36]. There is now much work demonstrating that hippocampal activity increases when sequential predictions are violated[37–40]. This increase has been localized to hippocampal area CA1 in both humans and rodents[38,41–43]. One interpretation of this increased CA1 BOLD signal during mnemonic prediction errors is that it may facilitate the encoding of the novel, unexpected, information, and thus promote memory updating and the improvement of future predictions[22,44]. Indeed, there is some behavioral evidence that mnemonic prediction errors facilitate episodic memory[38,45]. Critically, however, CA1 activation cannot speak to a shift in hippocampal states. As discussed above, these states are mediated by differential connectivity between hippocampal subfields, and not by CA1 univariate activity. We therefore set out examine whether mnemonic prediction errors are associated with a shift in hippocampal processing toward an encoding state that prioritizes input from the entorhinal cortex and away from a retrieval state[5,8,14,46]. Furthermore, we aimed to link these effects with the quality of the prediction itself.

To test these hypotheses, participants underwent extensive training to learn the furniture and layout of 30 distinct rooms. Then, in the fMRI scanner, we probed participants to retrieve each learned room by presenting a verbal cue (e.g., Johnsons boy's bedroom), which was then followed by a room image that either matched the learned room image or included changes (Fig. 1a). We operationalized the retrieval of the image as a form of memory prediction, and prediction errors were cases when the presented perceptual image was a violation of the actual learned image. Using high-resolution imaging, we find that mnemonic prediction errors bias CA1 functional connectivity toward entorhinal cortex and away from subregion CA3. Moreover, CA1 activity patterns during the cue show evidence of room-specific retrieval, potentially reflecting prediction of the learned room. Further, introducing changes in the room image leads to lower similarity between CA1 activity patterns during memory retrieval at the cue and during viewing the room image, consistent with a mnemonic prediction-error signal in CA1. Taken together, these findings show that mnemonic prediction errors bias CA1 functional connectivity, potentially to shift hippocampal processing to favor encoding and down-weight retrieval.

## Results

**Behavior.** A full reporting of the behavioral results has been provided in Duncan et al.[43], and is summarized here following a brief description of the task. We had two types of change-detection task: a Furniture task and a Layout task, in which participants indicated whether a change occurred in the identity or the layout of the furniture, correspondingly. On each trial, the room image included 0–2 task-relevant changes and 0–2 irrelevant changes. For example, in the Furniture task, there can be two task-relevant changes in the identity of the furniture, and one task-irrelevant change in the layout of the furniture (see Methods). As reported in Duncan et al.[43], a two (Task) by three (Relevant changes) by three (Irrelevant changes) repeated-measures ANOVA revealed that participants were more accurate in the Layout task compared with the Furniture task. Relevant changes did not interact with Task, however, introducing irrelevant changes did reduce accuracy in the Furniture task more than in the Layout task. Finally, relevant and irrelevant changes interacted, such that having no irrelevant changes increased accuracy, but only if there were no relevant changes as well (for more details, see Duncan et al.[43]). However, despite some

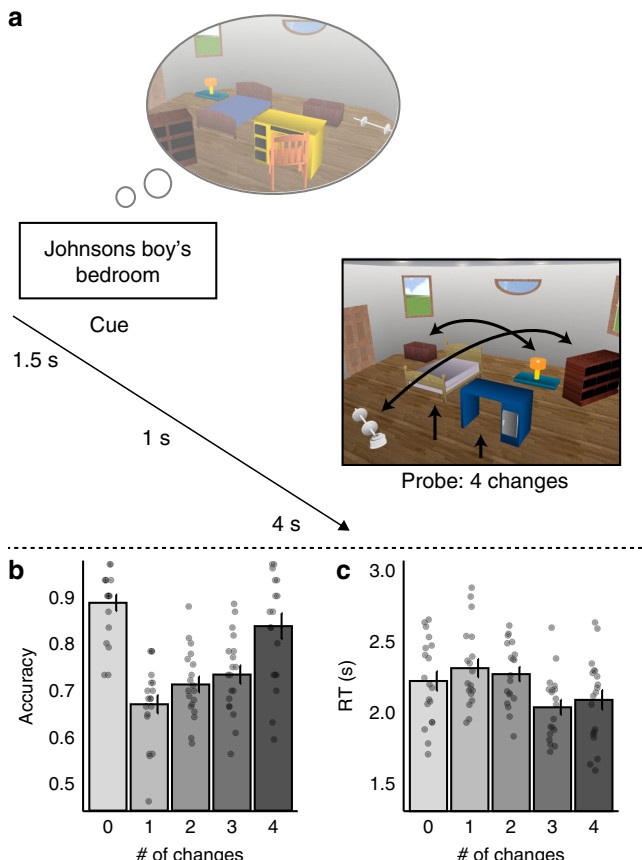

**Fig. 1 Trial example and behavioral results. a** Trial example: participants were presented with a cue probing them to retrieve a room image that they had extensively learned prior to the scan. After a short delay, they saw a probe image that included 0–4 changes relative to the learned image (four changes here), and indicated whether the seen image matched the learned image (see Methods). **b** Accuracy and **c** reaction times (RTs) in the match task. $N = 19$. Data are presented as mean values, error bars reflect $+/-$ SEM. Source data are provided as a Source Data file.

differences in behavioral effects of irrelevant and relevant changes, CA1 BOLD response predominately tracked the total number of changes, irrespective of relevance to the task[43]. Thus, in subsequent analyses, we collapse across relevant and irrelevant changes and report the behavioral and neural data as a function of the total number of changes. Accuracy data in the change-detection tasks were entered to a five (Changes: 0–4) by two (Task: Furniture/Layout) repeated-measures ANOVA. This ANOVA revealed main effects of Changes and Task, as well as an interaction (Changes: $F_{(4,72)} = 33.48$, $P < 0.0001$; Task: $F_{(1,18)} = 8.50$, $P = 0.009$; Interaction: $F_{(4,72)} = 3.24$, $P = 0.017$). In both tasks, accuracy was highest when there was no change (0 change) and in the 4-changes conditions in comparison with the 1- to 3-changes conditions.

Response times (RTs) also tracked the accuracy data: RTs were significantly shorter in the 0-changes and the 4-changes conditions compared with the 1- to 3 changes. These results reflect the relative ease of indicating "match" when there were no changes at all, or "mismatch" when there were many changes which provides support for the rooms having been well learned. RTs were also entered into the same ANOVA as the accuracy data, which again revealed main effects of Changes and Task, and an interaction (Changes: $F_{(4,72)} = 12.60$, $P < 0.0001$; Task: $F_{(1,18)} = 6.04$, $P = 0.024$; Interaction: $F_{(4,72)} = 7.66$, $P < 0.0001$). Mean and SD of accuracy and RT in each of the number of changes and each task

are provided in Table 1, and collapsed across tasks in Fig. 1. Importantly, in the neural data we did neither observe a main effect of Task nor an interaction between Task and Changes; thus, we collapsed across tasks (see Results).

**Mnemonic prediction errors bias CA1 functional connectivity.** Functional connectivity was measured using a beta-series correlation approach[47]. Since previous literature do not report consistent laterality of mnemonic prediction-error effects in the hippocampus[38–40,42,48], we looked at each hemisphere separately. Prior to testing our main hypothesis, we conducted, in each pair of anatomically defined ROIs, a five (Changes: 0–4) by two (Task: Furniture/Layout) repeated-measures ANOVA, to test whether collapsing across tasks is warranted. Indeed, there was neither main effect of Task nor a Changes by Task interaction in functional connectivity between CA1–CA3 (The CA3 ROI included CA2, CA3, and dentate gyrus) or CA1–entorhinal, for the left and the right hemispheres (all $P$'s > 0.17). Given this, we collapsed across tasks for our main analyses. First, functional connectivity was entered to a repeated-measures ANOVA with Hemisphere (right, left), ROI (CA3, entorhinal) and Changes (0–4) as independent variables. This ANOVA revealed a three-way interaction of ($F_{(4,72)} = 4.24$, $P = 0.004$, $\eta_p^2 = 0.19$), further justifying testing for the effect of levels of changes on differential connectivity between ROIs in each hemisphere separately. In the left hemisphere, we found an interaction between Changes (0–4) and ROI (entorhinal, CA3) using a repeated-measures ANOVA ($F_{(4,72)} = 6.04$, $P = 0.0003$, $\eta_p^2 = 0.25$), confirming our prediction that the number of changes in the presented room differentially modulated CA1 connectivity with the entorhinal cortex and area CA3 (Fig. 2). The same ANOVA conducted on the right hemisphere did not reveal a significant interaction ($P = 0.68$; no main effect of Changes, $P = 0.97$; a main effect of ROI was observed, $P = 0.001$). As the significant three-way interaction reported above qualified the specificity of the interaction between ROIs and Changes to the left hemisphere, further analyses were restricted to the left hemisphere ROIs.

Having established that the number of changes differentially modulated connectivity in CA1 pathways, we moved on to examine the connectivity of CA1 with each region (entorhinal, CA3) separately. As predicted, CA1–entorhinal connectivity increased with more changes (Fig. 2). By contrast, and again consistent with our predictions, CA1–CA3 connectivity decreased as number of changes increased (Fig. 2). One-way repeated-measures ANOVAs with the factor of Changes (0–4) conducted for each pair of regions separately, confirmed that in both region pairs (CA1–entorhinal or CA1–CA3), Changes significantly modulated connectivity (CA1–entorhinal: $F_{(4,72)} = 4.49$, $P = 0.0027$, $\eta_p^2 = 0.20$; CA1–CA3: $F_{(4,72)} = 3.58$, $P = 0.01$, $\eta_p^2 = 0.17$).

Although not the main aim of this study, we sought to further characterize the observed connectivity changes. To that end, we asked, for each pair of ROIs (CA1–entorhinal/CA1–CA3), whether connectivity changes correspond more to a linear trend, or rather to a simpler match < mismatch pattern. For each pair of ROIs, we constructed a mixed-level model, in which functional connectivity was the explained variable. As explaining variables, we included both a linear-trend contrast, in which the number of changes (0–4) were coded as linearly increasing numbers, and a match < mismatch contrast, in which the 0-change condition (i.e., match to the learned image) was compared with the 1–4-changes conditions grouped together, treating all trials with any change identically (see Methods). We then compared this full model to either a model including only the linear-trend contrast, or only the match < mismatch contrast. In CA1–entorhinal connectivity, we found that the full model significantly outperformed the linear

**Table 1 Accuracy rates and reaction times (for accurate responses) in the Layout and Furniture tasks.**

| Accuracy | 0 change | 1 change | 2 changes | 3 changes | 4 changes |
|---|---|---|---|---|---|
| Layout task | 0.89 (0.10) | 0.69 (0.09) | 0.74 (0.08) | 0.75 (0.10) | 0.90 (0.11) |
| Furniture task | 0.88 (0.10) | 0.65 (0.10) | 0.68 (0.10) | 0.72 (0.11) | 0.77 (0.17) |

| Reaction times | 0 change | 1 change | 2 changes | 3 changes | 4 changes |
|---|---|---|---|---|---|
| Layout task | 2.23 (0.33) | 2.28 (0.29) | 2.25 (0.22) | 2.06 (0.26) | 1.95 (0.36) |
| Furniture task | 2.22 (0.29) | 2.35 (0.28) | 2.30 (0.25) | 2.02 (0.26) | 2.24 (0.32) |

Reaction times are in seconds. SDs are in parentheses. Source data are provided as a Source Data file.

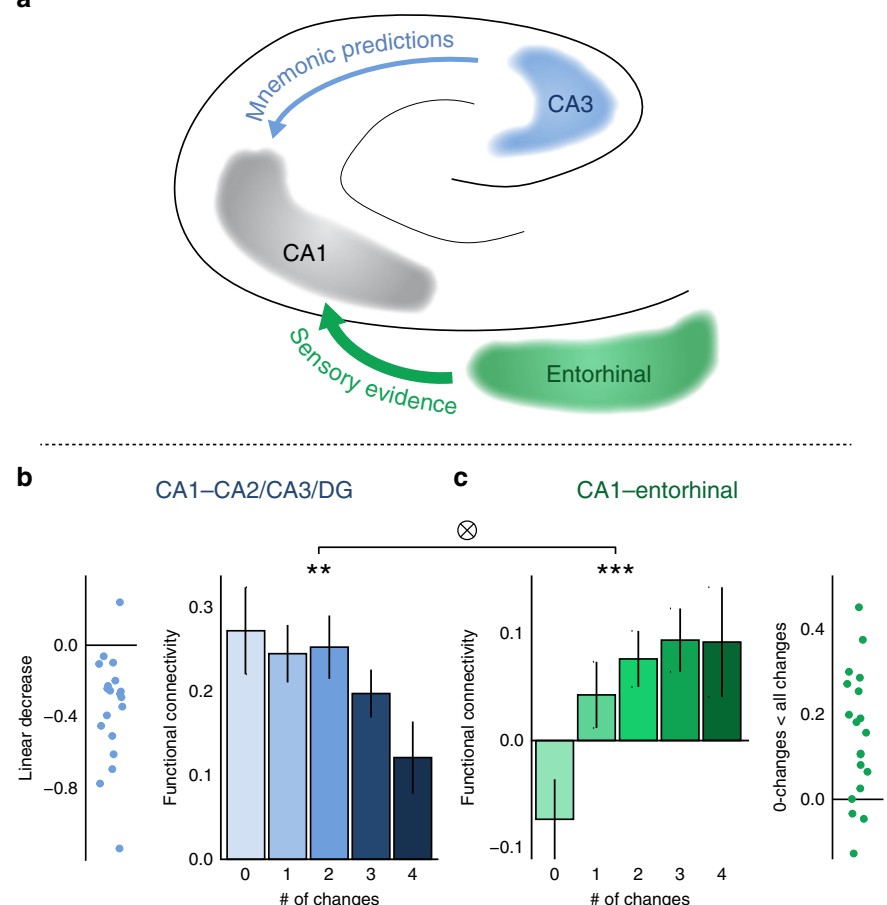

**Fig. 2 Functional connectivity with region CA1. a** Mnemonic prediction errors decreased CA1–CA3 interaction, while increasing CA1–entorhinal cortex interaction, potentially reflecting reduced processing of erroneous predictions, and upregulating processing of sensory evidence. **b** Functional connectivity of CA1 with region CA2/CA3/DG (blue). Dots in the dots plot reflect individual participants' linear decrease contrast score, computed across all number of changes (the preferable contrast in this pair of regions, main text). **c** Functional connectivity of CA1 with entorhinal cortex (green). Dots in the dots plot reflect individual participants' 0-changes (match) < average of 1–4 changes (mismatch; the preferable contrast in this pair of regions, main text). # of changes: number of changes. F-transformed beta-series correlation was our measure of functional connectivity. Data are from the left hemisphere (see main text). $N = 19$. Data in the bar graphs are presented as mean values, error bars reflect $+/-$ SEM. ⊗ Interaction of # of changes by ROI in a repeated-measures ANOVA: $P = 0.0003$. **$P = 0.01$, ***$P = 0.003$, the results of a one-way repeated-measures ANOVA within each pair of ROIs. Source data are provided as a Source Data file.

model ($\chi^2 = 4.39$, $P = 0.036$; AIC full: $-80.68$, linear: $-78.29$; BIC full: $-67.91$, linear: $-68.07$), but not the match < mismatch model ($\chi^2 = 1.31$, $P = 0.25$), suggesting that the match < mismatch contrast better describes CA1–entorhinal connectivity. For CA1–CA3 connectivity, the full mode significantly outperformed the match < mismatch model ($\chi^2 = 8.63$, $P = 0.0033$; AIC full: $-72.56$, match < mismatch: $-65.93$; BIC full: $-59.80$, match < mismatch: $-55.71$), but not the linear model ($\chi^2 = 0.59$, $P = 0.44$), suggesting that CA1–CA3 connectivity may decrease linearly as number of changes increase.

We further examined how many individual participants demonstrated the trends reported at the group level. We computed, for each participant, the match < mismatch contrast in CA1–entorhinal connectivity and the linear contrast in

CA1–CA3 connectivity (contrasts are as described above and in the Methods), as these were the contrast revealed to best explain variance in functional connectivity in these regions. Indeed, the overwhelming majority of the participants showed a positive match < mismatch trend in CA1–entorhinal connectivity (16 out of 19 participants, 84% of the participants), and a negative linear trend in CA1–CA3 connectivity (18 out of 19 participants, 95% of the participants).

We sought to control for univariate effects in the left CA1, CA3, and entorhinal cortex to exclude the possibility that univariate activation in these regions might have driven the connectivity findings. To that aim, we first evaluated the differences in univariate activation between levels of changes (0–4). In CA1, we found a significant linear increase in CA1 activation ($t_{(18)} = 2.43$, $P = 0.026$, Cohen's $d = 0.55$, CI: [0.0044–0.061]; linear-trend analysis, see Methods), reproducing the results reported by Duncan et al.[43]. Since CA1 connectivity showed opposite effects with CA3 vs. entorhinal cortex, we find it unlikely that CA1 univariate activation could account for the connectivity findings. We did not find any univariate differences between levels of changes in CA3 or in entorhinal cortex (linear trend: $t_{(18)}$'s < 0. 88, $P$'s > 0.39; simple effects between levels of changes: all $t_{(18)}$'s < 0. 1.5, $P$'s > 0.15; see also Supplementary Fig. 1). Thus, univariate activation in these regions is also unlikely to account for the differences we did observe in connectivity.

Nonetheless, we conducted a control analysis, whereby for each regions-pair (CA1–CA3/CA1–entorhinal), we included univariate activation in a mixed-level model together with the match < mismatch contrast or the linear-trend contrast as explaining variables, and connectivity as the explained variable (see Methods). We then compared these full models to a model including only univariate activation. For CA1–entorhinal connectivity, we took the match < mismatch contrast as our contrast of interest in the full model, since this was the preferable contrast that explained more variance in CA1–entorhinal connectivity. Indeed, the full model significantly explained more variance, and was preferable to the model including only the univariate activity ($\chi^2 = 14.32$, $P = 0.0002$, AIC or BIC reductions > 9.5). For CA1–CA3 connectivity, we used the linear contrast as our contrast of interest, since the linear contrast explained more variance in connectivity. Again, the full model including the linear contrast and univariate activity was preferable over the model with only univariate activity ($\chi^2 = 11.26$, $P = 0.0008$, AIC or BIC reductions >6.5). These results show that for both CA1–entorhinal and CA1–CA3 connectivity, the levels of changes significantly explain variance when controlling for univariate activation. Together, these analyses suggest that our main connectivity findings are unlikely to be explained by univariate activity. We further statistically controlled for accuracy and reaction times differences between participants and levels of changes, and our results held (Supplementary Note 2).

**CA1 activity patterns show room-specific predictions**. In the previous analysis, mnemonic prediction errors were operationalized as changes in the probe room image, relative to a retrieved memory of that room. Here, we sought to support the notion that our task involved mnemonic prediction errors, by providing evidence for room-specific predictions in CA1 during the cue, that is, when participants were asked to retrieve the room. To that end, we assessed the strength of the prediction in CA1, as estimated by the level of neural pattern similarity between a retrieved memory for a room compared with viewing of the same room. Specifically, prediction strength was estimated by correlating the multivariate BOLD activity pattern in CA1 during the presentation of each cue (e.g., "Johnson's boy's

bedroom", to which participants were instructed to retrieve a memory of that room) with the activity pattern measured when participants actually viewed the same room (the 0-changes image of the corresponding room) and comparing it to the correlation with the pattern evoked by 0-changes images of other rooms. Thus, the strength of mnemonic prediction should be reflected by the degree to which cue periods (when memories are generated) are more correlated with viewing the same as compared to other rooms. This analysis was restricted to the left hemisphere, where we had already obtained significant connectivity differences with the number of changes (Fig. 2). We found that the correlation with the corresponding room was higher than with the other rooms, suggesting that specific-room reinstatement took place in the left CA1 (match: $M = 0.01$, $SD = 0.01$; other: $M = 0.0045$, $SD = 0.005$; $t_{(18)} = 2.50$, $p = 0.022$; Cohen's $d$: 0.57, CI of the difference: [0.001–0.011]; 14 out of 19 participants, which are 74% of our participants, demonstrated qualitatively higher same- vs. other-rooms reinstatement; this result holds when controlling for univariate activation in CA1, see Supplementary Note 3). Interestingly, we have also found a moderate correlation across participants between prediction strength in CA1 and the increase in CA1–entorhinal connectivity in response to errors (see Supplementary Fig. 2 and the related Supplementary Note 1), lending further support for the notion that functional connectivity increases are related to predictions and their violations.

**CA1 activity patterns reflect mnemonic prediction errors**. The previous result suggests that CA1 activity patterns during the cue capture participants' predictions. However, it does not directly examine participants' prediction errors. Here, we estimated a mnemonic prediction-error signal in region CA1 by measuring the difference between participants' multivoxel activity patterns during the cue (i.e., the mnemonic prediction) and during the violations. To assess the level of mnemonic prediction errors in CA1, we computed the correlation between the multivoxel activity patterns of the prediction during the memory cue and the violation when viewing the room in the same trial. First, correlation values were submitted to a repeated-measures ANOVA, with Changes (0–4) and Task (Furniture/Layout) as within-participant factors. Since no interaction was obtained ($F_{(4,72)} = 0.34$, $P = 0.85$), we collapsed across tasks for further analyses. We found that pattern similarity in CA1 decreased as the number of changes increased (see Fig. 3). Specifically, the match > mismatch contrast revealed to be highly significant (contrast $M = 0.01$, $SD = 0.01$, CI: [0.003–0.015], $t_{(18)} = 3.14$, $P = 0.006$, Cohen's $d = 0.72$; 14 out of 19 participants, which are 74% of our participants, demonstrated a positive match > mismatch contrast; Source data are provided as a Source Data file). This result further holds when controlling for univariate activation (Supplementary Note 4), and importantly, when subtracting the similarity of the cue to images of the same trial type but of other rooms, suggesting that this decrease in similarity reflects room-specific representation rather than some broader decrease in pattern similarity in response to mismatches (Supplementary Note 5). Further, the match > mismatch contrast seemed to characterize the decrease better than the linear contrast, which was only marginally significant (contrast $M = 0.005$, $SD = 0.014$, CI: [−0.001–0.012], $t_{(18)} = 1.76$, $P = 0.09$, Cohen's $d = 0.40$, see Fig. 3). CA1 activity patterns thus are sensitive to the mismatch between a retrieved memory and perceptual input that is an altered version of that memory.

## Discussion

Behavioral and physiological work have implicated hippocampal processing in both laying down new memories and retrieving past memories[9–14]. The computational principles that underlie these

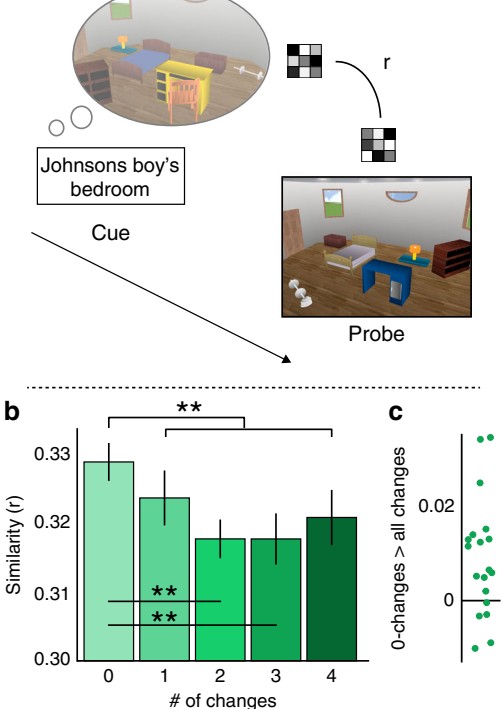

**a**

Johnsons boy's bedroom

Cue

r

Probe

**b**

**c**

**Fig. 3 Mnemonic prediction errors in CA1. a** Mnemonic prediction error was assessed by computing the pattern similarity between the cue and the probe parts of the trial. **b** CA1 similarity between the cue and the image decreased when changes were introduced in the images. Data in the bar graphs are presented as mean values, error bars reflect $+/-$ SEM. $**P <$ 0.01. The result depicted by the black lines above the bars reflect the 0-changes (match) > average of 1–4-changes (mismatch) contrast, tested against 0 with a one-sample two-tailed $t$ test. The other significance marks reflect a paired-sample $t$ test between 0-changes and a specific level of changes. **c** Dots in the dots plot reflect individual participants' scores in the 0-changes > average of 1–4-changes contrast. $N = 19$. Source data are provided as a Source Data file.

processes are in conflict as encoding will benefit most from synaptic plasticity, while, during retrieval, plasticity may alter the memory trace and lead to inaccurate memory representations[6,14,15,17]. To address this apparent conundrum, it has been proposed that encoding and retrieval may be mediated by distinct hippocampal states[5,8,13,18,32,49]. Specifically, recent work has linked functional coupling between CA1 and the entorhinal or perirhinal cortices with encoding and CA1–CA3 coupling with retrieval operations[5,19,20,25,29,30,33–35,50].

Here, we leveraged these findings to ask whether interactions between internal memory states and conflicting environmental evidence can dynamically modulate or bias hippocampal processing states in predictable ways. To the extent that violations of expectations drive new learning or encoding, they should adaptively bias CA1 processing of inputs from medial temporal lobe (MTL) cortical regions. At the same time, these mnemonic prediction errors might down-weight projections from the now incorrect memory-based predictions from CA3 to CA1. To test that hypothesis, participants were cued to retrieve previously well-learned images of rooms. Memory retrieval was then followed by the visual presentation of images that either matched or mismatched the learned information (see Methods and Results). Consistent with our hypothesis, we found that in the left hemisphere, CA1 connectivity with the entorhinal cortex increased as mnemonic prediction errors increased. This was accompanied by

a decrease in CA1–CA3 connectivity for those same trials. Thus, mnemonic prediction errors do not simply lead to an overall general increase (or decrease) in functional connectivity of the CA1 region, but rather they selectively and differentially modulate processing along distinct hippocampal pathways.

To support the notion that connectivity changes were related to participants' internal memory predictions, we quantified prediction strength and mnemonic prediction error by examining the multivoxel activity patterns in CA1. We found higher similarity between activity patterns corresponding a retrieved memory of a room and viewing of that same room compared with other rooms, indicating room-specific memory reinstatement, or prediction, in CA1. Further, as a supplemental finding (Supplementary Fig. 2), we see evidence that participants with better cued memory reinstatement showed a greater increase in CA1–entorhinal connectivity in response to subsequent violations of the remembered rooms. This across-participants correlation provides initial evidence for the notion that CA1–entorhinal connectivity is related to CA1 prediction strength. It would be fruitful to further examine this relationship in a within-subjects design. In addition, because the cue and probe aspects of each trial were temporally proximal in our design (which was done to maximize the effect of a mnemonic prediction error), it would be beneficial to measure memory strength in temporally separate trials. We have additionally found that in CA1, activity patterns during cued memory reinstatement were more similar to activity patterns during viewing the same image, compared with viewing an altered version of image. This result is consistent with the notion that CA1 activity patterns are sensitive to mnemonic prediction errors. Together, our results suggest that an interplay between internal memory predictions and environmental evidence modulate hippocampal processing states, potentially driving hippocampal processing towards an encoding state and away from a retrieval state[6,8,14,18,36]. Such state shifts may prove to be an adaptive mechanism for memory updating: by reducing processing of erroneous retrieved predictions while upregulating encoding of the novel sensory evidence.

How the hippocampus shifts between memory states is largely unknown. It is possible that both acetylcholine (Ach) and dopamine (DA) play a role in biasing hippocampal states[4,8,51–53]. Some models propose that novelty detection in the hippocampus upregulates Ach input, which in turn increases excitation in the CA1–entorhinal pathway, while dampening CA1–CA3 communication[8,34]. It has also been proposed that Ach input may further entrain theta and gamma frequencies associated with encoding versus retrieval states[8,19,34,52,54]. Another influential theory suggests that increased CA1 activity in response to prediction errors leads to an increase in activation in the ventral tegmental area (VTA), a primary source of DA, which in turn projects back to CA1 and entorhinal cortex[53]. Supporting evidence comes from fMRI studies showing concomitant hippocampal and VTA activation in response to novel and unexpected events[55,56], and VTA–CA1 interactions were recently shown to mediate associative memory encoding[57] (see ref. [58] for review). In rodents, injection of DA agonist to the CA1–entorhinal pathway increased the CA1 post-synaptic potential, suggesting that DA can increase CA1–entorhinal synaptic transmission[59] (cf. ref. [60]). Thus, it is possible that CA1 activation leads to engagement of the postulated back-projection from VTA to CA1 and entorhinal cortex[53] and serves to functionally couple these regions and enhance CA1–entorhinal connectivity. Consistent with that notion, we show preliminary results suggesting that connectivity in CA1–entorhinal cortex was correlated with the strength of the memory predictions measured in area CA1. Namely, participants who showed greater similarity between a viewed room and the rooms' retrieval cue, our measure of a mnemonic prediction,

exhibited larger increases in CA1–entorhinal connectivity in response to presented rooms that contained changes, or violations, of the learned room. This interpretation of synchronization via VTA input to CA1 and entorhinal cortex is further consistent with the lack of a correlation between CA1–CA3 connectivity and CA1 prediction strength (Supplementary Fig. 2). More work is needed, however, to better understand how neurotransmitters such as DA, Ach, and potentially norepinephrine[51,61,62] contribute to a shift in hippocampal connectivity with changing mnemonic demands.

Another intriguing question for further research is whether and how functional connectivity between hippocampal subregions is related to pattern completion versus pattern separation mechanisms[10,14–17]. Pattern completion, namely, the reinstatement of a previously encoded activity pattern, has been suggested to underlie memory retrieval[16,63–65]. Pattern separation, or the allocation of a distinct activity pattern to similar experiences, is suggested to underlie encoding of new memories[66–69]. Likewise, previous empirical work has provided evidence that communication between CA1 and CA3, and between CA1 and entorhinal cortex may support retrieval versus encoding, correspondingly[20,25,33,35,50]. This study, however, did not aim to directly test whether pattern completion or separation took place as a result of mnemonic prediction errors, but to examine whether memory violations are related to changes in intrahippocampal connectivity. Thus, the specific computations mediated by communication between hippocampal subregions or by the subregions themselves in the case of mnemonic prediction errors, are exciting questions for future research to explore.

One limitation of this study is that our CA3 ROI included the CA2 subregion and the dentate gyrus (DG). While this limitation is necessary given the resolution of our data and is shared by many human studies of hippocampal subfields[70–72], recent advances in functional imaging such as 7-Tesla MRI scanners, might enable researchers to distinguish between CA3 and DG[67]. Relatedly, we note that we based our manual segmentation on a T1-weighted image, while some recent proposals recommend using an additional T2-weighted image[73]. The main advantage in using a T2 image is in distinguishing the corno ammonis (CA) from dentate gyrus (DG)[73,74]. As mentioned above, we did not aim to distinguish CA3 from DG in this study. Our manual segmentation was performed based on a well-established procedure relying on markers clearly visible on a T1 image[75,76], thus we believe that this concern is unlikely to influence our results. Importantly, we report that functional connectivity between hippocampal subfields was modulated in opposite directions: CA1–CA3 connectivity increased, while CA1–entorhinal cortex connectivity increased, as number of changes in the images increased. Any possible blurring of ROIs, if occurred, would therefore impair our ability to observe this dissociation. Future work will be able to further specify the reported connectivity findings.

While the accounts discussed above place the CA1 region as the source of violation-detection and connectivity changes[6–8,13,38,43,53,77], it is possible that prediction errors are also detected in earlier brain regions. Experimental and computational work in the predictive coding framework converge on the notion that high-level areas project top-down predictions to earlier visual cortices, where these predictions are then compared to incoming sensory information[78–80]. Consistent with this, after learning that a stimulus predicts another visual stimulus, greater activity was reported in visual cortex of both humans and monkeys in response to stimuli that violated such memory-based predictions, compared to stimuli that confirmed prior expectations[81,82] (for reviews, see refs. [83,84]). Moreover, it is now widely reported that memory reinstatement in cortical regions is correlated with hippocampal activity[85–91]. While fMRI studies cannot resolve the temporality of neural activity, a recent ECoG study found that memory reinstatement in visual processing regions preceded hippocampal reinstatement in humans[92]. Together, these studies suggest that memory reinstatement, or predictions, may occur in early processing stages, and hence then influence subsequent hippocampal processing. Like memory predictions, it is also possible that early prediction-error signals as those mentioned above may propagate forward to influence hippocampal processing[44] and potentially mediate connectivity changes.

A critical assumption in models of CA1 function is that CA1 may be ideally suited to compare internal memory output with input from visual cortical regions representing ongoing visual experience[6,7,53,93]. While earlier investigations have reported increased BOLD signal during mnemonic prediction errors in the hippocampus and, specifically, in area CA1[38–40,43], these studies neither specifically measure memory predictions in CA1 nor could they address the content of CA1 processing. Thus, whether the content of CA1 processing indeed reflects predictions as well as incoming sensory input, or whether univariate findings reflect other violation-related processes remained unknown. Here, we found that in CA1, activity patterns during cued memory reinstatement were more similar to activity patterns during viewing the same image, compared to viewing an altered version of image (see Results, Fig. 3). This result suggests that CA1 multivoxel pattern representations are sensitive to the difference between internal memory representations and sensory evidence, thus providing essential evidence to support the role of CA1 as a violation detector[6,7,53,77,93].

In summary, we found that mnemonic prediction errors biased hippocampal area CA1 connectivity toward entorhinal cortex and away from area CA3. We propose that this bias may reflect a shift in hippocampal states toward encoding of the novel sensory information and away from retrieval of erroneous memory-based predictions. How the hippocampus supports both encoding and retrieval is an intriguing question that has received increased attention in recent years[4,18,32,46,64]. The current results contribute to this ongoing line of research by measuring hippocampal states in humans, and by suggesting that the interplay between memory reinstatement as a prediction and their subsequent violation, or mnemonic prediction errors, may be an important factor in biasing these states. Thus, in addition to understanding the distinct neural mechanisms that allow shifting between encoding and retrieval, future research should aim at understanding the psychological factors that may shift our cognitive system between these different mnemonic states[4,8,13].

## Methods

**Participants**. Twenty participants were included in this study (mean age: 25.4 years). Further information can be found in Duncan et al.[43], where the results of univariate analyses of these data were previously published. One participant was removed from all analyses due to substantial entorhinal dropout (see Regions of interest). As was mentioned in Duncan et al.[43], all participants provided informed consent in a manner approved by the Institutional Review Board at New York University.

**Procedure**. In the training phase (~24 h prior to scanning, and again before entering the scanner), participants were extensively trained to identify each of 30 named rooms (e.g., "Johnson's boy bedroom") to criteria[43]. While scanning, participants were employed in two change-detection tasks. In both tasks, the room's name appeared for 1.5 s, followed by 1-s blank and a probe image (4 s). The probe image contained 0–2 changes in the individual pieces of furniture, along with 0–2 changes in the layout of the furniture, relative to the learned image, making a total of 0–4 changes per image. In the Furniture task, participants were asked to indicate whether all pieces of furniture were identical to the studied image. In the Layout task, participants were asked to indicate whether the layout of the furniture was identical to the learned image. This resulted in a two (Task: Furniture/Layout) by five (Changes: 0–4 total changes) within-participant design. Each room appeared once in every trial type (nine trial types: 0/1/2 furniture changes by 0/1/2 layout

changes), across both tasks, to make a total of 270 trials. Here, we focused on total number of changes (0–4 total changes, see below). Thus, analysis was conducted on 30 trials in each of the 0 and 4 changes, 60 trials in the 1- and 3-changes conditions, and 90 trials in the 2-changes condition (across both tasks). Tasks were blocked, such that each scan included one task (ten scans, five per task), and the blocks alternated between the Furniture and the Layout task. One participant had eight blocks, and another had seven. The minimal number of trials per condition was 24 and 21, correspondingly, still allowing a meaningful analysis. Hence, these participants were included in the analysis.

**FMRI parameters**. Scanning was performed using a 3-T Siemens Allegra MRI system. A high-resolution EPI sequence was used to collect functional data (TR = 2.5 s, TE = 49 ms, FOV = 192 × 96 mm, 26 interleaved slices, distance factor of 20%, 1.5 × 1.5 × 2 mm voxel size). A T1-weighted high-resolution MPRAGE (1 × 1 × 1 mm voxel size) was used as an anatomical scan.

**Regions of interest (ROIs)**. Anatomical ROIs were drawn manually by K.D. on each participant's MPRAGE anatomical image, and were then registered to functional space. The same hippocampal ROIs (CA1, CA2/CA3/DG) reported in Duncan et al.[43] were used here. These ROIs were drawn in a similar procedure to Kirwan et al.[75]. The entorhinal cortex was drawn using guidelines discussed in Insausti et al.[94] and Pruessner et al.[95]. ROIs were also masked to remove voxels with substantial signal dropout, a concern mainly in the entorhinal cortex[96]. One participant with only 12 voxels in the left entorhinal and 80 voxels in the right entorhinal was excluded from all analyses. All other participants had on average 234 voxels in the left entorhinal ROI (range: 127–344), comprising 84% (range: 44–93%) of the anatomical left entorhinal. In the right entorhinal ROI, participants averaged 255 (range: 165–337) voxels, which were 87% (59–95%) of the anatomical right entorhinal.

**Functional connectivity**. *fMRI beta-series correlation*. Functional connectivity between regions was computed using a beta-series correlation approach[47], in which a timeseries of single-trial parameter estimates in two regions are correlated. To obtain the single-trial estimates, we used an LSS (least-square-separate) approach[97–99]. We reasoned that this approach would maximize our ability to capture the variance explained by the image portion of each trial (our focus of interest) and distinguish this variance from preceding cue part of each trial (the name of each room). Thus, in the first level analysis, a separate GLM was computed for each trial. Each model included the image portion of a single trial as a regressor of interest. The cue portion in all trials were included in one regressor of no interest. Other images were binned based on trial type to make nine additional regressors of no interest. In all regressors, events were modeled as boxcars lasting for the duration of the event (1.5 s for cues, 4 s for images) convolved with a double-gamma function to approximate the hemodynamic response. A temporal derivative regressor was also added for each regressor. GLMs were implemented using FSL FEAT. This procedure yielded 270 parameter estimates, one for each trial. A t-stat was computed for each parameter estimate, and these were averaged, per each trial, across all voxels in each ROI (CA1, CA2/CA3/DG, entorhinal cortex, and perirhinal cortex, separately for right and left hemispheres). T-stats were then binned based on experimental conditions: number of changes (0–4) and task (Furniture/Layout) to make 10 t-series for each ROI. We then computed functional connectivity between area CA1 and the other brain regions of interest: CA2/3/DG and entorhinal cortex in each of the ten conditions separately for each hemisphere. The Pearson's r values per each participant, condition, and pair of ROIs were Fisher-transformed and entered to the group-level analysis.

We further aimed to control for univariate effects in CA1, CA3, and entorhinal cortex, to exclude the possibility that univariate activation might have driven any connectivity findings. To that aim, for each participant, we averaged the t-statistics corresponding to activation estimates in these same trials that were used to compute connectivity, based on the level of changes (0–4). The statistical procedure we used to evaluate univariate effects and to control for such potential effects is detailed below in the section dedicated to statistical procedures. For the functional connectivity and univariate analyses, as well as for the analyses below, all the analysis steps after obtaining the t-statistics were performed using a costume code in MATLAB R2018b (The MathWorks Inc) or in R (version 3.5.2; R Core Team, 2018), where mentioned. More details can be found in the documentation at https://github.com/odedbein/Ticky_public, where all the costume code is available.

**CA1 mnemonic prediction strength analysis**. In order to measure the strength of participants' mnemonic predictions, we used a representational similarity analysis[100,101] (RSA). To obtain the multivoxel activity pattern for each cue, we used the same LSS procedure as for the images (see Functional connectivity: beta-series correlation). Each cue was allocated a separate GLM, which included one regressor of interest for the cue, and a few regressors of no interest: one regressor for all other cues, and nine additional regressors modeling the images—one for every trial type. As with the image models, a time-derivative regressor was added for each regressor. Parameter estimates were then converted to t-statistics, which were taken to the RSA.

To boost SNR for the representational similarity analysis, we removed noisy/non-responsive voxels by eliminating a third of the voxels that were least activated

by our task as determined by an independent GLM[68,102,103]. Specifically, we conducted a GLM that included a regressor per trial type (nine trial types, see above) capturing the image part of the trial, as well as one regressor for all cues (as the models were conducted in the run level, and tasks were divided between runs, this analysis effectively is conducted for each task separately). Events were boxcars lasting for the duration of the event (1.5 s for cues, 4 s for images) and convolved with a double-gamma function to approximate the hemodynamic response. A temporal derivative regressor was also added for each regressor. We then computed per-participant average activation for images per each level of changes (0–4), in each task, and averaged across the level of changes and tasks, to get the average activation level in our task. Then, for each participant, we excluded a third of the left CA1 voxels that were least activated by our task from the RSA analyses. The results without voxel selection were largely consistent and are reported in the Supplementary Note 6.

To compute the strength of participants' mnemonic predictions, we correlated the multivoxel activity pattern in CA1 observed in response to each room cue with the multivoxel activity pattern measured when participants viewed the intact room image (i.e., the 0-changes image). For example, the CA1 activity pattern in response to the verbal cue "Johnsons boy's bedroom" was correlated with the CA1 activity in response to the intact image of Johnsons boy's bedroom. To compute the similarity to the specific match image, while controlling for condition-level effects and general similarity to all 0-changes images, we computed, for each cue, the correlation between the activity pattern during the cue and the activity pattern of other 0-changes images, and averaged across these correlation values. Then, we subtracted this average correlation with other 0-changes images from the correlation with the intact image corresponding to the cue (e.g., the intact image of Johnsons boy's bedroom). This yielded, for each cue, a measure of how good the prediction of the specific corresponding room was, beyond overall similarity to a 0-changes image. This procedure further controlled for differences in average similarity values between participants, which is critical for a meaningful interpretation of across participant correlations of prediction strength with connectivity. Cues in some trials were excluded from this analysis: first, we excluded cues in the 0-changes condition. These cues were presented in the same trial as the corresponding intact image while all other 0-changes images were presented in other trials, thus we avoided comparing within-trial similarity to across-trial similarity. Second, we excluded cues and intact images that were presented in the same scan to avoid inflating similarity values within the same scan[97]. Third, we only took cues in which the cue and the intact image were presented in the same task, to avoid introducing task differences between the cue and the image. For each participant, the correlation values between the cues that entered the analysis and their corresponding 0-changes images (other 0-changes images subtracted, as detailed above) were averaged and Fisher-transformed per task and then averaged across both tasks to obtain a prediction index per participant.

**CA1 multivariate mnemonic prediction-error analysis**. To further support our hypothesis that mnemonic prediction errors modulate hippocampal connectivity, we aimed to compute a measure of mnemonic prediction error in our study. To this end, we correlated the CA1 activity pattern (same voxel selection as for the prediction strength analysis) during the presentation of each cue when participants were instructed to retrieve a memory of the cued room (i.e., the mnemonic prediction) with the CA1 activity pattern measured when viewing the probe image on each trial (the sensory evidence). We reasoned that the difference between the representation of the mnemonic prediction and that of the sensory evidence can be interpreted as a mnemonic prediction error. We averaged this value across all the trials within each number of changes (0–4), and separately in each task, and Fisher-transformed these correlation values for statistical analysis. If indeed participants retrieved the intact image on each trial, we predicted a decrease in similarity, or increased prediction error, as number of changes increased, reflecting larger divergence between the retrieved memory and the sensory evidence.

**Statistical tests for the functional connectivity analysis**. In the group-level analysis of the functional connectivity data (beta-series correlation), Fisher-transformed r values in each pair of ROIs were entered to a five (Changes: 0–4) by two (Task: Furniture/Layout) repeated-measures ANOVA. We saw no interaction between Task and Changes in CA1 connectivity with either CA3 or entorhinal cortex. Thus, for CA1–CA3 and CA1–entorhinal, for each participant in each number of changes, we collapsed across tasks to obtain an average beta-series correlation value. Since previous univariate findings do not show consistent lateralization[39,40,42,48] (see also ref. [104]), we reasoned to look at each hemisphere separately. Thus, we first conducted an ANOVA of Hemisphere (left/right) by Changes (0–4) by ROI (CA3 vs. entorhinal). To preview, this analysis indeed revealed a three-way interaction of Hemisphere by Changes by ROI (see Results). We thus tested our hypotheses regrading a shift in CA1 connectivity in each hemisphere separately. To directly test our main hypothesis that mnemonic prediction errors modulate CA1 connectivity with CA3 vs. entorhinal cortex, we conducted a five (Changes: 0–4) by 2 (ROI: CA3 vs. entorhinal) repeated-measures ANOVA in each hemisphere. Where a Changes by ROI interaction was observed, we tested how Changes (0–4) influenced connectivity separately in each pair of ROIs (CA1–CA3, CA1–entorhinal), using a one-way repeated-measures ANOVA.

Although we had no specific hypothesis regarding the shape of the increase or decrease in connectivity, we sought to further characterize connectivity changes. We asked whether connectivity changed linearly with number of changes, or, alternatively, whether changes may reflect a binary match–mismatch signal, whereby any level of change is different from no changes at all, with no or little difference between level of changes. To that end, we defined a linear contrast by allocating for each number of changes (0,1,2,3,4) linear-trend values (−2, −1, 0, 1, 2) correspondingly. The match < mismatch contrast was defined as by coding the 0-changes condition as −1, whereas the 1–4-changes conditions were coded 0.25 each. We directly compared the linear-trend contrast to the match < mismatch contrast by using a mixed-effects model approach as implemented by lmer function in R[105]. We included both contrasts as explanatory variables in the same model (the beta-series correlation value per participant per number of changes was the explained variable) and then compared this full model to either a model including only the linear-trend contrast, or only the match < mismatch contrast (match < mismatch was treated as a factor, an intercept per participant was included in all models). This analysis thus examines whether one contrast significantly explains variance above and beyond the other contrast.

We further sought to control for univariate effects in regions CA1, CA3, and entorhinal cortex. To that end, we first evaluated whether there were any differences in univariate activation between level of changes, aiming to reproduce the results reported in the original publication of these data[43]. Thus, we followed the same approach as in that previous study[43], and conducted a linear-trend analysis, whereby a linear contrast is computed per participant (by coding level of changes as −2, −1, 0, 1, 2) and these contrast scores are tested against zero at the group level using one-sample two-tailed $t$ test. We then further estimated simple effects between level of changes using paired-sample two-tailed $t$ tests.

To statistically control for any univariate effects in our main connectivity analysis, we adopted a mixed-level modeling approach. This control analysis was only conducted in the left hemisphere, were we found connectivity effects (see Results). For each regions-pair (CA1–CA3/CA1–entorhinal), a model was constructed (using lmer function in R[105]), including connectivity in each level of changes per each participant as our explained variable. As explaining variables, we included univariate activity in each level of changes per participant for both regions of the pair, as well as their interaction. As our explaining variable of interest, for the left CA1–entorhinal connectivity, we took either the linear contrast, or the match < mismatch contrast, as both explained variance in connectivity between these regions (although the match < mismatch was preferable). In CA1–CA3, we only took the linear contrast, as the match < mismatch contrast did not significantly explain variance in the main analysis. A random intercept was included per participant in all models. Model comparison was used to estimate the models: we compared a full model, with the contrast of interest and univariate activity as detailed above, to a model that included only the univariate activity. If the full models including our contrast of interest and univariate activation explain significantly more variance than the models including only univariate activation, we can conclude that our connectivity findings are unlikely to be attributed to univariate activation differences.

**Statistical tests for the prediction strength and mnemonic prediction-error analyses**. The significance of the prediction strength (namely, room-specific memory reinstatement: the difference between the similarity of the cue to the corresponding intact room image, and the similarity of the cue to intact room images of other rooms) was tested with a paired-sample two-tailed $t$ test.

For the mnemonic prediction-error analysis, we first entered the Fisher-transformed similarity values to a five (Changes: 0–4) by two (Task: Furniture/ Layout) repeated-measures ANOVA. To preview, since there was no interaction between Changes and Task in CA1, we collapsed across Task in all further analyses. Like in the functional connectivity analysis, we then estimated this decrease using a linear-trend analysis, as well as a match < mismatch analysis. We calculated a contrast score per participant using the same contrasts as described above, and tested these contrasts against zero using a two-tailed, one-sample $t$ test.

**Reproducibility**. This is a single fMRI experiment. We did not repeat the experiment and no replication attempts have been made to date.

**Reporting summary**. Further information on research design is available in the Nature Research Reporting Summary linked to this article.

## Data availability

Minimally processed data and single-trial t-statistic maps that support the findings of this study, along with the stimuli used in the task, are available online (https://osf.io/re2wd/). A reporting summary for this Article is available as a Supplementary Information file. Additional data are available from the corresponding author upon reasonable request. Source data are provided with this paper.

## Code availability

The costume code that was used for data analysis in this study is available at https://github.com/odedbein/Ticky_public.

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

## Acknowledgements

This research was supported by the National Institute of Mental Health Grant R01MH074692 to L.D. O.B. is further supported by a McCracken fellowship.

## Author contributions

O.B., L.D., and KD conceptualized the general experimental design. O.B. and L.D. conceptualized the specific analytic approach reported in this paper. O.B. analyzed the data. K.D. conducted the preprocessing of fMRI data and regions of interest demarcation. O.B., L.D., and K.D. wrote the paper. K.D. and L.D. conceptualized and designed the task. K.D. collected the data.

## Competing interests

The authors declare no competing interests.
