## [Peer Review File · Nature Communications]

Reviewers' comments:

Reviewer #1 (Remarks to the Author):

Bien and colleagues report a re-analysis of a previously described data set that uses a memory paradigm in which subjects learn the names of various rooms and then engage in a task, during fMRI scanning, where a room name (cue) appears followed by a probe that consists either of an intact version of the room, or a slightly modified version of the room. Previously, it was reported that CA1 was sensitive to changes (mismatch) in the probe rooms, relative to the original rooms. Here, the authors look at connectivity between CA1 and entorhinal cortex and between CA1 and CA2/3/DG as a function of whether or not changes occurred. The key finding is that left CA1-EC connectivity increases as a function of the number of changes whereas left CA1-CA2/3/DG connectivity decreases as a function of the number of changes. Additionally it is argued that subjects that exhibit stronger reinstatement in CA1 during the cue period show greater increases in CA1-EC connectivity when mismatches occur.

The paper is well written and the ideas are very nicely motivated from human and rodent literatures. The idea that mismatch trials shift the 'state' of the hippocampus toward greater connectivity with EC is interesting and the result would be of potentially broad relevance. The idea that reactivation strength plays a role in this shift is also a potentially noteworthy finding. That said, my enthusiasm for the findings decreased as I made my way through the details of the results. The first result appears relatively 'solid' but it is restricted to left hippocampus regions, for reasons that are not discussed. Additionally, I would characterize this result as a somewhat incremental advance on the previously-reported findings (which already emphasized the role of CA1 in signaling mismatches). The second result (related to reactivation strength), is much less convincing (for multiple reasons, which I detail below). Some of the other analyses also suffer from weaknesses that raise questions about the interpretations. Thus, despite some initial excitement for the ideas in the paper, I was not sufficiently convinced that the paper delivered on its promise.

Comments

1. Previously, the authors reported (using the same data set) that univariate responses in CA1 were modulated by the number of changes (again, in a linear manner). However, I don't believe that univariate responses in entorhinal cortex were reported. Does EC show similar univariate responses to CA1? If so, is the connectivity between CA1-EC simply a consequence of both regions exhibiting similar sensitivities to changes?

2. There is no overall (significant) evidence of reinstatement. The logic of the between subject correlation is that some subjects do/might show more reinstatement than others. But, this is a non-optimal approach (trying to find variance in an effect that is not independently established). Thus, the correlation only provides very indirect evidence to support the idea that reactivation occurred. Additionally, between-subject analyses are less compelling than within-subject analyses (due to the number of things that can co-vary across subjects); and this is compounded by the low n for this study (less than 20 subjects).

3. Related to the above point: the temporal offset between the cue and the probe is only 2.5s, which raises questions about how/whether the cue and probe period can be cleanly separated. I realize that for the cue prediction analyses there was a separate regressor for the cue and probe parts of the trial, but just because these were separately modeled does not mean that variance was cleanly partitioned. The between subjects correlation between CA1 prediction strength and CA1-entorhinal connectivity is based on pattern similarity between cue activity and probe activity, but to the extent that the cue

regressor picks up some variance related to the probe, then this could reduce to pattern similarity between (matching) probe images. There is no great way to address this because it is a limitation of the design. And, as noted above, the between subject correlation is already underwhelming given that (a) there is no overall reactivation effect, (b) the sample size is low, and (c) it's a one-tailed test. That said, at a minimum it would be worth repeating the analysis but this time deliberately correlating matching – mismatching probe regressors. If the cue-probe correlation is ultimately just picking up on probe-probe similarity, then deliberately using a measure of probe-probe similarity should NOT yield the same correlation. Or, even better, probe-probe similarity could be regressed out of the cue-probe similarity measure so that cue-probe similarity is providing statistically independent information. But, as is, the overall evidence supporting the idea that reactivation occurred or that reactivation triggers greater CA1-entorhinal connectivity is very shaky.

4. The analysis in Figure 4 is interesting, but there needs to be a control comparison between cues and probes that don't correspond to the same room. The reason being that the lower correlation with more changes could just reflect a generic shift in activity patterns when changes occur, as opposed to a real prediction of the image.

5. Are the results in Figure 4 related to the judgments (memory decisions) subjects ultimately make? It seems that similarity might be lower to the extent that subjects view the probe as different. Combined with the above point, this analysis could potentially be recast as showing lower similarity between retrieval (cue) and encoding (when probes are judged to be 'new') without it reflecting anything about room-specific predictions. While I think this analysis is somewhat distinct from the other, main analyses, the authors do draw strong conclusions from this analysis: "This result suggests that the content of CA1 representations are sensitive to the difference between internal memory representations and sensory evidence, thus providing essential evidence to support the role of CA1 as a violation detector."

6. The abstract should explicitly note that the observed effects were specific to left hemisphere regions. Because the key effects were not present (at all) in the right hemisphere, it's misleading to describe the results without respect to hemisphere in the abstract. The selectivity of the results to the left hemisphere also warrants a reminder in the Discussion.

7. For the data shown in Figure 2 (line 198 in text), an ANOVA is reported which is presumably the main effect of # of changes, and this is described as testing whether there was a significant increase or decrease in connectivity with the number of changes, but the ANOVA just shows that connectivity varies across conditions (which is a useful point, but slightly different from how the authors frame the result). The mixed effects model described later does test the linear trend, so the ANOVA should just be described more neutrally.

Reviewer #2 (Remarks to the Author):

The manuscript seeks to understand how pattern completion/separation might relate to unexpected feedback during retrieval of recently learned room layouts or objects within rooms. The authors sought to test a model developed in rodents that suggests that ERC-CA1 interactions might be important to encoding (pattern separation) and CA3-CA1 connectivity important to retrieval (pattern completion). The authors sought to test this model by having subjects learn rooms, which they then retrieved in the scanner. The subjects received feedback that either matched or mismatched the cue

by 1-4 features. The authors found that (left) ERC-CA1 connectivity increased with mismatches and (left) CA3-CA1 connectivity decreased with mismatches. The correlation between the putative reinstated trace and the actual room was not different from the correlation for mismatched rooms. CA1 multivariate pattern differences for each subject as a function of room correlated with ERC-CA1 connectivity although there was no negative correlation for CA3-CA1. There was also a tendency of mismatched features to be lower for CA1 multivariate patterns than the same room. The authors conclude that gating by different paths within the hippocampus (ERC-CA1 vs. CA3-CA1) may be important for mediating pattern completion and pattern separation dynamically during tasks.

The patterns of findings are interesting here and testing the models presented, even if their exact mapping in the task is slightly unclear, is potentially useful. There are some concerns, however, with the results and statistical differences/consistency with past work on this topic. The theoretical motivations also appear somewhat unclear in places. The MRI methods may also be suboptimal in some cases. Aspects may be of interest to a broad audience, although other aspects seemed better suited for an audience with experience in memory. I detail some of these concerns below.

MAJOR

1) Given some of the past work on the topic, it seems important to replicate that basic finding that univariate findings differ as a function of match vs. mismatch (Duncan, Ketz, Inati, & Davachi, 2012). Did this effect replicate? If so, how did (or did not) univariate pattern relate to multivariate patterns reported here? It is also notable that the hippocampal multivariate effect (Figure 4) does show decreases in pattern similarity but not in the linear fashion reported in some studies (Lacy, Yassa, Stark, Muftuler, & Stark, 2010; Stokes, Kyle, & Ekstrom, 2015). Even here, the match-mismatch effect is relatively weak ($p < .04$). A searchlight analysis could potentially help better resolve if there is a signal to noise issue here with the CA1 ROI or potentially something also present in CA3 (see point 2). While ERC-CA1 connectivity showed a positive trend and CA3-CA1 showed a negative trend, were these slopes different? It is also notable that CA3-CA1 connectivity did not correlate with CA1 multivariate pattern differences (unlike ERC-CA1). This might have been expected if CA3-CA1 connectivity was mediating the effect but was not found, suggesting that ERC-CA1 (and not CA3-CA1) input matters to CA1 neural patterns. More generally, given that the authors did not find CA3-CA1 connectivity correlations with CA1 multivariate patterns, it seems important to try to better understand why not. It is also notable that the correlation with the reinstated room and the actual room image did not differ significantly (line 247). This is somewhat of a concern because if CA1 contains information about the trace and its mismatch, this effect should likely have been there. The authors attribute this to variance but other studies have found such effects, at least in the hippocampus (Mack & Preston, 2016). Thus, it was surprising not to see this in the data in some form. Finally, the authors use one-tailed t-tests in many places. These should be avoided as they have a high risk of false positives.

2) The authors reported using standard MPRAGE anatomical sequences (line 453) and then co-registering a high-resolution EPI to this sequence. However, there are reasons one might expect the MPRAGE to give insufficient signal to noise and contrast to identify the subfields. For one, T2 TSE are considered the gold standard for identifying subfields and an MPRAGE is unlikely to have the differences in contrast/signal needed to discriminate CA1 from CA3/DG on most slices (Wisse et al., 2016). The concern is that at least some of the weak/non-significant effects could be driven by poor subfield identification. The authors should consider a searchlight analysis. Ideally, a T2 image would be available for more precise segmentations.

3) Theoretical: The authors base the ideas of the study on a model developed and tested largely (but not exclusively) in rodents. However, one issue is that the single neuron/LFP signals studies by groups

like the Colgin lab don't have a clear mapping onto the BOLD signal, which is likely a mix of input from different subfields (Logothetis, 2003). While testing the model is reasonable, it is not clear that what the authors are observing with connectivity is an actual up-weighting of one pathway and down-weighting of the other. This is accentuated by the fact that CA3-CA1 connectivity did not correlate with CA1 multivariate patterns. As such, it is difficult to link the findings precisely to the model mentioned in the introduction. In addition, pattern completion and separation are almost certainly happening during both encoding and retrieval. So for example, retrieval is likely to involve both separation (separating the memory from the other interfering/competing ones) and completion to the desired output. This is likely to happen very rapidly and it seems unclear whether BOLD can completely capture these given the relatively poor temporal resolution. Of note, the design nicely attempts to separate these aspects, however, it is not clear the results can directly contact this aspect of the Hasselmo/Colgin models.

MINOR

1) The rationale for the findings being in left hemisphere was a little unclear. The authors should be sure to test full models with hemisphere as a factor first and then conduct post-hoc analyses accordingly. In addition, one might have expected right sided activation given the spatial nature of the retrieval component, although such lateralities are inconsistent with fMRI.

2) ""In CA1-entorhinal connectivity, we found that the full model significantly outperformed the linear model ($\chi^2 = 4.39$, $p < .05$), but not the match < mismatch model ($\chi^2 = 1.31$, $p > .25$), suggesting that the match < mismatch contrast better describes CA1-entorhinal connectivity."

Ideally, AIC/BIC scores could be used here to deal with potential differences in degrees of freedom, unless these are identical?

References

Duncan, K., Ketz, N., Inati, S. J., & Davachi, L. (2012). Evidence for Area CA1 as a Match/Mismatch Detector:

A High-Resolution fMRI Study of the Human Hippocampus. *Hippocampus*.

Lacy, J. W., Yassa, M. A., Stark, S. M., Muftuler, L. T., & Stark, C. E. (2010). Distinct pattern separation related transfer functions in human CA3/dentate and CA1 revealed using high-resolution fMRI and variable mnemonic similarity. *Learn Mem*, 18(1), 15-18.

Logothetis, N. K. (2003). The underpinnings of the BOLD functional magnetic resonance imaging signal. *J Neurosci*, 23(10), 3963-3971.

Mack, M. L., & Preston, A. R. (2016). Decisions about the past are guided by reinstatement of specific memories in the hippocampus and perirhinal cortex. *Neuroimage*, 127, 144-157.

Stokes, J., Kyle, C., & Ekstrom, A. D. (2015). Complementary roles of human hippocampal subfields in differentiation and integration of spatial context. *J Cogn Neurosci*, 27(3), 546-559.

Wisse, L. E., Daugherty, A. M., Olsen, R. K., Berron, D., Carr, V. A., Stark, C. E., et al. (2016). A harmonized segmentation protocol for hippocampal and parahippocampal LID - 10.1002/hipo.22671 [doi]. *Hippocampus*.

Reviewer #3 (Remarks to the Author):

The authors present work demonstrating differential functional connectivity between CA1 and CA3/DG and Entorhinal cortex during a scene change detection task, and evidence that this reflects a change in the network state dependent on prediction errors for the scenes.

Overall, I greatly enjoyed reading this manuscript. It was clear and well-written, and the motivation for the work is well founded. Analyses were generally laid out in a very principled manner, with one typically addressing alternative interpretations left over from the preceding analysis.

The results are original and of potentially high impact.

I found myself with only one category of related comments and concerns, which should be straightforward to address:

1) There are several allusions to parametric relationships with room similarity in the manuscript, usually presented as a follow-up test on a significant ANOVA effect of # of changes. It would seem the range of these tests was guided by the authors' predictions (i.e., change/no change vs linear), which is great to see and easy to follow – but the ANOVA and visual inspection of the data leave open the possibility that the relationships actually take other untested forms.

Specifically, when scrutinizing the behavioral data, one could walk away with two impressions about the change manipulation: from an accuracy perspective, it appears change levels 1-3 are fundamentally different from 0 and 4. From the RT data it seems as though participants' default strategies are to assume that there is a change in the scene and they search until they rule this out or confirm – and interestingly, only with 3-4 changes does the perceptual mismatch map onto a change in speed. Both of these suggest a more step-wise influence of the experimental manipulation (although at somewhat different change levels for ACC vs RT). Could it be the case that participants move from a serial search to a more holistic scene matching process at different levels of change? Likewise, it's a bit tricky to think about when mismatches are actually going to be well-represented in the neural data (i.e., is there a sharp boundary at 2-3 or 3-4?).

a. It would be helpful to know the simple effects here between levels (and in the subsequent neural analyses across change levels)

b. Would the authors expect a more clear mismatch signal for their neural data if only examining the subset of trials for 1-4 where participants were correct (indicating they did ultimately detect the change)? If this was a flat distribution, and different from the one reported in which the hits and misses are combined in the data (if I understood the methods correctly), this would lend further support to the gradient in the reported data being driven by the proportion of trials where the mismatch was detected.

2) Largely the same comment, but applied to the functional data, I found it interesting that the CA1-CA3/DG pattern – although tested as linear or match-mismatch – really appears to exhibit a gradient starting at 2-3 changes (not unlike the behavioral steps occurring in that range). Here it would again be nice to know the simple effects coming out of the significant ANOVAs, and I began to wonder if an alternative “change/no change” contrast motivated by the behavior rather than the design would better explain the relationship between CA1-CA3/DG. There has been work mapping decision thresholds and memory judgment confidence to different components of the declarative memory

system in episodic memory tasks (e.g. <https://www.ncbi.nlm.nih.gov/pubmed/23019246>, with some evidence linking these concepts to MTL structures as well when searching semantically-familiar scenes for personally-relevant episodic cues - <https://www.nature.com/articles/s41598-018-24549-y>) and I wonder if such a behaviorally-grounded analysis strategy, combined with the authors' between-subjects correlation analysis, could give more leverage over the response profiles across change level.

3) The pattern similarity analyses are a very nice addition and important to the arguments being made. They did leave me wondering about the univariate activity data. To what extent does trial-trial variance change across the different change bins in the data (relevant to the connectivity analyses) and to what extent does CA1 activity change from 0-4? For example, if CA1 activity decreases with the amount of mismatch in the task this could color the attribution of the pattern similarity data prediction errors (e.g., univariate signal is known to influence similarity amplitudes).

Reviewers' comments:

Reviewer #1 (Remarks to the Author):

Bein and colleagues report a re-analysis of a previously described data set that uses a memory paradigm in which subjects learn the names of various rooms and then engage in a task, during fMRI scanning, where a room name (cue) appears followed by a probe that consists either of an intact version of the room, or a slightly modified version of the room. Previously, it was reported that CA1 was sensitive to changes (mismatch) in the probe rooms, relative to the original rooms. Here, the authors look at connectivity between CA1 and entorhinal cortex and between CA1 and CA2/3/DG as a function of whether or not changes occurred. The key finding is that left CA1-EC connectivity increases as a function of the number of changes whereas left CA1-CA2/3/DG connectivity decreases as a function of the number of changes. Additionally it is argued that subjects that exhibit stronger reinstatement in CA1 during the cue period show greater increases in CA1-EC connectivity when mismatches occur.

The paper is well written and the ideas are very nicely motivated from human and rodent literatures. The idea that mismatch trials shift the 'state' of the hippocampus toward greater connectivity with EC is interesting and the result would be of potentially broad relevance. The idea that reactivation strength plays a role in this shift is also a potentially noteworthy finding.

We want to thank the reviewer for this positive assessment of the novelty of our findings.

That said, my enthusiasm for the findings decreased as I made my way through the details of the results. The first result appears relatively 'solid' but it is restricted to left hippocampus regions, for reasons that are not discussed. Additionally, I would characterize this result as a somewhat incremental advance on the previously-reported findings (which already emphasized the role of CA1 in signaling mismatches). The second result (related to reactivation strength), is much less convincing (for multiple reasons, which I detail below). Some of the other analyses also suffer from weaknesses that raise questions about the interpretations. Thus, despite some initial excitement for the ideas in the paper, I was not sufficiently convinced that the paper delivered on its promise.

We thank the reviewer for first emphasizing the broad implications of our work, namely, that mismatches shift hippocampal states. However, later the reviewer evaluates the main finding as 'somewhat incremental in light of previous reported findings 'which already emphasized the role of CA1 in signaling mismatches'. Because connectivity results (correlated activity between two areas) is an independent measure from activation in one region, we do not agree that these findings are in any way incremental. Importantly, our results show that CA1, a region that is sensitive to changes in expectations, exhibits both

increases and decreases in connectivity with different regions. We think this latter finding makes the point very strongly that the univariate CA1 activation effects reported in prior studies do not necessarily mean one can draw inferences about connectivity effects (see supporting analyses below). Importantly, it is exactly these connectivity effects, and not CA1 activation, that are proposed to mediate different hippocampal states.

We thank the reviewer for raising this possibility because it highlights the need for us to emphasize this point more in the paper. In the Introduction, we added the following sentence: “Critically, however, CA1 activation cannot speak to a shift in hippocampal states. As discussed above, these states are mediated by differential connectivity between hippocampal subfields, and not by CA1 univariate activity” (p. 5). We appreciate the reviewer comment that this point might have not come across clearly enough, and we believe that this sentence clarifies better the critical addition of our connectivity findings to the growing and influential body of work on hippocampal states. We further answer this concern by statistically controlling for univariate activation using mixed-effects linear models, and showing that our connectivity results hold. We provide detailed explanation of these analyses, and address the concern about the laterality of our findings, after the reviewer’s detailed comments below.

1. Previously, the authors reported (using the same data set) that univariate responses in CA1 were modulated by the number of changes (again, in a linear manner). However, I don’t believe that univariate responses in entorhinal cortex were reported. Does EC show similar univariate responses to CA1? If so, is the connectivity between CA1-EC simply a consequence of both regions exhibiting similar sensitivities to changes?

We thank the reviewer for bringing up this important concern. Indeed, there is an increase in CA1 activity in response to changes that was previously reported in Duncan et al. 2012. As we now report (p.10), we reproduced this linear increase. However, and as we now also report, there is no difference in activation based on number of changes in either entorhinal cortex (EC) or in CA3. Thus, the increased connectivity between CA1-EC and decreased connectivity between CA1-CA3 cannot be explained by univariate changes in these areas. Nevertheless, to respond more directly to the concern about univariate activation and to establish that the connectivity findings are independent of univariate activation findings, we have also now run an additional control analysis: for each pair of regions (CA1-EC/CA1-CA3) we adopted a mixed-effects linear model with connectivity in each level of changes as the explained variable, and as explaining variables we included the average univariate activation per participant in each of the regions in the pair (i.e., CA1 and EC or CA1 and CA3), as well as the interaction of univariate

activation. In addition to the univariate activation, as our main contrast of interest we included in the model as an explaining variable either the linear trend contrast (0-4 level of changes were coded as -2,1,0,1,2) or the match < mismatch contrast (0-change was coded as -4, whereas 1-4 level of changes coded as 1). As a reminder, we found that for CA1-EC connectivity, the match < mismatch contrast better accounted for the pattern of connectivity changes thus we included the match < mismatch contrast as our contrast of interest. For CA1-CA3, a linear contrast better accounted for the connectivity changes, thus we included the linear trend contrast as our contrast of interest. Including univariate activation in our models together with these contrasts (i.e., match < mismatch/linear trend) allowed us to test whether the differences in connectivity across levels of changes are significant also when statistically accounting for univariate activation. In these models, the results remain consistent. Specifically, we compared these full models including both our contrast of interest and univariate activity to models that only included univariate activity (additionally, an intercept per participant was included in all models). We found that for both pairs of regions, the models with the contrast of interest significantly explained more variance compared to the models including only univariate activity (CA1-EC: a model including the match < mismatch contrast: $\chi^2 = 14.32$, $p < .001$, AIC or BIC reductions > 10; CA1-CA3: a model including the linear contrast: $\chi^2 = 11.26$, $p < .001$, AIC or BIC reductions > 6). These results show that for both CA1-EC and CA1-CA3 connectivity, the levels of changes significantly explain variance when controlling for univariate activation. Thus, they suggest that our main connectivity findings are unlikely to be explained by univariate activity in any of the three regions involved. We thank the reviewer as these additional analyses strengthen our findings, and we now report the univariate activity as well as the additional control analyses in the manuscript (Results, pp. 10-11).

2. There is no overall (significant) evidence of reinstatement. The logic of the between subject correlation is that some subjects do/might show more reinstatement than others. But, this is a non-optimal approach (trying to find variance in an effect that is not independently established). Thus, the correlation only provides very indirect evidence to support the idea that reactivation occurred. Additionally, between-subject analyses are less compelling than within-subject analyses (due to the number of things that can co-vary across subjects); and this is compounded by the low n for this study (less than 20 subjects).

We thank the reviewer for this comment. We wanted to point out, first, that there is so far scant evidence for hippocampal memory reinstatement in the literature, highlighting that while it has been shown (Mack et al., 2016; Tomparry et al., 2016), it has so far only been noted in high-resolution studies in hippocampus and,

thus, is likely a noisy effect. We also do not agree that variance in a measure should not be considered informative when lacking a main effect. In fact, when there is a main effect, it may be precisely in cases when variance is low and thus examining variance around a main effect in those cases may be less informative, or optimal.

Nonetheless, putting this aside, in line with a suggestion of Reviewer #2 and to potentially boost the SNR, we excluded a third of the voxels with the lowest univariate activation (voxel exclusion was done per participant in the left CA1, where we report reinstatement effects). This approach of removing low-activation voxels has previously been used in multivariate representational analyses (Favila et al., 2016; Chanals et al., 2017). We then computed room-specific reinstatement by correlating, for each cue, the multivoxel activity pattern during the cue with the activity pattern of the match image (the 0-changes image) of the corresponding room, and averaged across cues. We then compared that same-room similarity to the similarity of the cue to match images corresponding to other rooms, to obtain a measure of room-specific reinstatement (as was done in the original manuscript). In the left CA1, the qualitative difference we have had previously between the similarity to the corresponding room image compared to other room images has now reached statistical significance ($t_{(18)} = 2.5$, $p = .01$, one-tailed; the reasons for using one-tailed t-test are detailed below, but note that the result is significant using two-tailed test as well, $p = .02$). We now report this analysis in the Results (p. 13). For completeness, we report the results without voxel selection in the Supplementary Information.

We further find that this result holds when controlling for univariate activation during both the cue and the intact images. Specifically, we computed, for each participant, the average activation during the presentation of cues as well as during the presentation of match images that were used in the analysis (note that in the reinstatement analysis we excluded some of the cues and images to control for task differences and to avoid mixing within trial and across-trials similarity, as well as to avoid computing similarity of trials within the same scan, see Methods, pp. 26-27. Thus, in this control analysis, to control for univariate activation as closely and accurately as possible, we only included univariate activation from trials that were used for the main reinstatement analysis). Then, we included in a multiple regression the room-specific reinstatement measure per participant (namely, the difference between same-room similarity and other-rooms similarity) as the explained variable, and the average univariate activation per participant during the presentation of the cues and of the images. The intercept of the model thus corresponds to the magnitude of the reinstatement,

when accounting for univariate activation. Indeed, the intercept in this multiple regression was significant ($t_{(16)} = 2.41, p < .02$), suggesting that reinstatement cannot be explained by univariate activation (Supplementary Information).

We also agree with the reviewer that across-subject correlations are less compelling than within-subject effects. In this case, for this specific analysis, we were interested in following up on our main result showing connectivity changes with CA1 to see if those connectivity changes were related to reactivation measures. However, it is important to note that our ability to obtain a within-subject measure was limited since the connectivity measure is a measure of correlation across all trials – making it one measure per condition per participant. Additionally, the reviewer questioned whether the effects we see with an across-participant correlation may reflect trait level differences across participants. To that end, it is important to note that each data point in the across-subjects correlation reflects a within-subjects subtraction, and not some overall measure of brain activity. Specifically, each data point reflects the difference in measures of reinstatement to the same room versus different rooms (within participants) and the change in connectivity across levels of change (again, within participants). Thus, because the across-participant correlation is made up of subtractions of measures within participants, we would argue that it is less likely to reflect trait level differences across participants. We now explain this logic more clearly in the results section, as we introduce the correlation: “Thus, note that we are correlating two within-participant measures: (1) The similarity of the cue to its own matching room, compared to the similarity of the cue to other rooms (i.e., we are computing room-specific reinstatement, and subtract from that “baseline” similarity of the cue to rooms more broadly). (2) The increase in connectivity in response to changes in the images, namely, the match < mismatch contrast score per participant. This more conservative approach to across participants correlation ensures that we are not correlating two baseline and potentially trait-level brain measures, but rather a within-participant measure of the strength of room-specific reinstatement, with the increase in connectivity in response to errors” (pp. 13-14).

To further answer the reviewer’s concern, we now additionally control for univariate activation, both during the cue and during the image, and the correlation is still significant. Specifically, we conducted a multiple linear regression that included the match < mismatch connectivity contrast as explained variable, and as explaining variables we included the CA1 reinstatement measure, univariate activation in CA1 during the cue, and during the match images, as well as the match < mismatch univariate contrast in CA1 and in EC (to account for univariate activation for connectivity). Even when including all of these other factors, the reinstatement variables was nevertheless

significant ($t_{(13)} = 1.93, p < .05, \text{one-tailed}$). Thus, the correlation is unlikely explained by univariate activation differences between participants. Another concern is that outliers may sway the correlation, especially in small samples. However, as can be seen in the spread of our data, there were no such outliers. Nevertheless, to account for such a possibility, we used robust regression procedure, which accounts for outliers, and the correlation is still significant ($t_{(13)} = 1.83, p < .05, \text{one-tailed}$). We now report all of these additional analyses in the Supplementary Information.

Finally, we do agree that it is important to develop different experimental paradigms to allow for the measurement of trial to trial changes in reactivation as they relate to shifting connectivity over time, perhaps reflecting state changes in the hippocampus. Our hope is that our results will promote future studies to replicate and extend these initial findings using different paradigms. We have added this point in the Discussion: “This across-participants correlation provides initial evidence for the notion that CA1-entorhinal connectivity may be related to CA1 prediction strength. It would be fruitful to further examine this relationship in a within-subjects design” (pp. 18-19).

3. Related to the above point: the temporal offset between the cue and the probe is only 2.5s, which raises questions about how/whether the cue and probe period can be cleanly separated. I realize that for the cue prediction analyses there was a separate regressor for the cue and probe parts of the trial, but just because these were separately modeled does not mean that variance was cleanly partitioned. The between subjects correlation between CA1 prediction strength and CA1-entorhinal connectivity is based on pattern similarity between cue activity and probe activity, but to the extent that the cue regressor picks up some variance related to the probe, then this could reduce to pattern similarity between (matching) probe images. There is no great way to address this because it is a limitation of the design. And, as noted above, the between subject correlation is already underwhelming given that (a) there is no overall reactivation effect, (b) the sample size is low, and (c) it's a one-tailed test. That said, at a minimum it would be worth repeating the analysis but this time deliberately correlating matching – mismatching probe regressors. If the cue-probe correlation is ultimately just picking up on probe-probe similarity, then deliberately using a measure of probe-probe similarity should NOT yield the same correlation. Or, even better, probe-probe similarity could be regressed out of the cue-probe similarity measure so that cue-probe similarity is providing statistically independent information. But, as is, the overall evidence supporting the idea that reactivation occurred or that reactivation triggers greater CA1-entorhinal connectivity is very shaky.

We first want to clarify that similarity in this analysis was computed between the cues and probes across different trials. Nevertheless, we understand that the reviewer is concerned about bleeding of the pattern of the probes that

immediately follow the cues (in the same trial), to the pattern of the cue. To address the reviewer's concern, we have run the analysis suggested: we ran the same correlation only this time computing the similarity between the activity pattern of the probe and the activity pattern of the intact image as our measure of reinstatement (what the reviewer refers to as probe-probe similarity). If indeed our cue-match image correlation with connectivity is carried by bleed over from the probe, then the correlation between connectivity and the similarity between the probe-match image should be higher than the correlation we observed when taking cue-match image similarity. We did not see this. By contrast, we found that this correlation was actually lower ($r = .158$) than the correlation previously reported when using the cue ($r = .40$).

Importantly, we also note that, if the correlation reflects similarity of the probe to the intact (0-changes) image, then participants that had probes that were more similar to their match image (in their CA1 multivariate activity patterns) also had higher CA1-Entorhinal connectivity. However, and critically, our main effect in connectivity shows that more similarity (i.e., 0-changes condition vs. 1-4 level of changes) between the probe and the intact image led to lower CA1-entorhinal connectivity. Thus, we think it is unlikely that probe-probe similarity drives our correlation.

We also wanted to address the comment about using one-tailed tests. We respectfully disagree that one-tailed t-test makes our result weaker. One-tailed t-tests are justified when there are a priori directional hypotheses, rather than testing for any difference that allows the rejection of the null-hypothesis, as is the case in two-tailed tests (Ruxton and Nauhauser, 2010). Of course, these a priori reasons need to be detailed so that they can be evaluated by readers. Here, for both the reinstatement analysis, and the correlation with connectivity - we outline the priors that led us to embrace one-tailed statistical tests.

Specifically, for the reinstatement measure:

- 1. Prior research has shown memory reinstatement in CA1 (Mack et al., 2016; Tomparry et al., 2016).***
- 2. We computed the difference between similarity of the cue to the intact image of the same room (e.g., the cue of Johnson's boy bedroom with the intact image of Johnson's boy bedroom) to similarity of the cue to other rooms (e.g., the cue of Johnson's boy bedroom with the intact image of the Smith's living room). Clearly, the cue should be more similar to the intact image of the same room compared to others, and there is no reason to believe the other direction (more similarity to other images) would occur.***

In any case, we further note that the reinstatement measure is also significant using a two-tailed t-test.

Regarding the correlation measure:

The computational models that motivated our work (Hasselmo, 2002; Meeter 2004) provide strong theoretical foundations for a directional hypothesis. Specifically, they argue that more novelty - and specifically, stronger mnemonic prediction errors, should lead to a shift towards an encoding 'state' mediated by CA1-entorhinal connectivity. Thus, we hypothesized that prediction errors should specifically increase CA1-Erc connectivity while also decreasing CA1-CA3 connectivity - both are directional hypotheses and warrant one-tailed tests.

We agree with the reviewer that our reasoning might have not been laid out clearly enough in the manuscript and thus made it harder to evaluate the justification of one-tailed t-tests. We now further motivate the one-tailed tests and specify the reasons in the Methods section (p. 30) and thank the reviewer for raising this concern.

Although we feel there is worth in reporting the correlation between reinstatement and connectivity as a secondary analysis to our main connectivity finding, we also are sympathetic to the reviewers' concerns. Since this is the first empirical evidence that we are aware of that shows that mnemonic prediction errors modulate hippocampal connectivity, we think it is advantageous to report this analysis, in order to facilitate future research. We nevertheless acknowledge that the across-participants correlations, even well-controlled, may still be considered weaker evidence. Thus, we have significantly qualified our interpretation of that correlation in the manuscript. Specifically, we mention the across-participants nature of the correlation in the abstract: "Further, stronger memory predictions measured in CA1 during the cue correlated across participants with the CA1-entorhinal connectivity increase in response to violations" (p. 2). In the Discussion, we note: "This across-participants correlation provides initial evidence for the notion that CA1-entorhinal connectivity is related to CA1 prediction strength. It would be fruitful to further examine this relationship in a within-subjects design. Additionally, because the cue and probe aspects of each trial were temporally proximal in our design (which was done to maximize the effect of a mnemonic prediction error) it would be beneficial to measure memory strength in temporally separate trials" (pp. 18-19). Finally, we do not feel that this secondary analysis is the main strength of the paper and would be open to moving it to the Supplementary Information, if the reviewer think it would improve the paper.

4. The analysis in Figure 4 is interesting, but there needs to be a control comparison between cues and probes that don't correspond to the same room. The reason being that the lower correlation with more changes could just reflect a generic shift in activity patterns when changes occur, as opposed to a real prediction of the image.

We thank the reviewer for this comment and agree with the reviewer. We ran the requested additional control analysis, and the results were consistent. Specifically, for each cue, we subtracted from the similarity between the cue and the image of the corresponding room, the average similarity of that cue to all room images that are of the same trial type (task and number of changes; to closely control for condition-level differences), but importantly, that correspond to other rooms. This measure gives us a room-specific measure of cue-image similarity. Like in our original analysis, we found that the match > mismatch contrast was significant ($t_{(18)} = 2.60, p = .018$). Importantly, we should have not obtained this result if the patterns only reflected a generic shift in activation patterns.

We now mention this additional analysis in the manuscript “This result further holds when controlling for univariate activation, and importantly, when subtracting the similarity of the cue to images of the same trial type but of other rooms, suggesting that this decrease in similarity reflects room-specific representation rather than some broader decrease in pattern similarity in response to mismatches” (p. 16), and report it fully in the Supplementary Information. We want to thank the reviewer for this comment as we feel the addition of this analysis strengthens the paper.

5. Are the results in Figure 4 related to the judgments (memory decisions) subjects ultimately make? It seems that similarity might be lower to the extent that subjects view the probe as different. Combined with the above point, this analysis could potentially be recast as showing lower similarity between retrieval (cue) and encoding (when probes are judged to be ‘new’) without it reflecting anything about room-specific predictions. While I think this analysis is somewhat distinct from the other, main analyses, the authors do draw strong conclusions from this analysis: “This result suggests that the content of CA1 representations are sensitive to the difference between internal memory representations and sensory evidence, thus providing essential evidence to support the role of CA1 as a violation detector.”

We thank the reviewer for raising this concern. As noted above, we subtracted the similarity of the cue to the patterns of room images of the same trial-type, but that correspond to other rooms, and the match > mismatch contrast remained significant ($t_{(18)} = 2.60, p = .018$). Critically, these other probes contained the same level of changes, and thus engendered similar responses to the main image. Thus, if our results were driven by low similarity between “old” vs. “new” responses, subtracting other images should have diminished our effect, as other images are also “old”, or “new”. Nevertheless, our results hold, excluding this possibility.

We also examined the results from inaccurate responses separately to further examine the issues the reviewer raises. In the data reported in the initial manuscript, we included all trials. Importantly, for inaccurate responses, the response to the intact image is “new” (or mismatch), while the responses to changed images is “old” (or match). Thus, if our results reflect higher similarity of the retrieval cue to images determined as “old”, we might expect higher similarity between cues and images with changes, compared to match images, when looking at inaccurate responses. This is not the pattern we see: for inaccurate responses (see below; “match”/“mismatch” denote the response the participants were making), not only do we not see higher cue-image similarity for images with changes compared to match images, but in fact, we see the opposite: cue-image similarity for images with changes, that were deemed “old”, is still lower than cue-image similarity in for images with 0-changes that were deemed “new” (the match > mismatch even approached significance, $p < .07$, two-tailed, note that since we trained participants extensively, they were highly accurate, thus this analysis has less power, and nevertheless, we obtained lower similarity)

Thus, taken together, we think these two points argue that our similarity results do not simply reflect the similarity of the cue to distinct responses of “old” vs. “new” that participants make.

Nevertheless, we do not want to over interpret our results and have modified our interpretation of the data. Rather than mentioning the ‘content of CA1 responses’, we say that these are the multivoxel pattern representations, which is a description of the results that is closer to the actual data: “This result suggests that CA1 multivoxel pattern representations are sensitive to the difference between internal

memory representations and sensory evidence, thus providing essential evidence to support the role of CA1 as a violation detector.” (Discussion, p. 21)

6. The abstract should explicitly note that the observed effects were specific to left hemisphere regions. Because the key effects were not present (at all) in the right hemisphere, it's misleading to describe the results without respect to hemisphere in the abstract. The selectivity of the results to the left hemisphere also warrants a reminder in the Discussion.

We thank the reviewer for making this point. We now mention that specificity in the abstract: “We found that in the left hemisphere, CA1-entorhinal connectivity increased, and CA1-CA3 connectivity decreased, with the number of changes to the learned rooms.” (p. 2) and the Discussion: “Consistent with our hypothesis, we found that in the left hemisphere, CA1 connectivity with entorhinal cortex increased as mnemonic prediction errors increased. This was accompanied by a decrease in CA1-CA3 connectivity for those same trials” (p. 18).

We note that lateralization of hippocampal findings is common. Previous univariate mismatch findings do not show consistent lateralization. Specifically, while Dudukovich et al. (2011), as well as Kumaran et al. (2006,2007) found a prediction-error effect in the left CA1, Chen et al (2011, 2015) report this effect in a bilateral ROI (making it hard to know if there was a lateralization of the effect). Given this prior literature, we typically examine each hemisphere separately and entered the resulting data into an ANOVA testing for laterality effects, which indeed yielded an interaction level of changes, by ROI (CA1-entorhinal/CA1-CA3), by hemisphere (left/right). We now explain this rational in the Methods: “Since previous univariate findings do not show consistent lateralization (Chen et al. 2011, 2015; Dudukovich et al. 2011; Kumaran et al. 2006,2007; see also Jordan, 2019), we reasoned to look at each hemisphere separately. Thus, we first conducted an ANOVA of Hemisphere (left/right) by Changes (0-4) by ROI (CA3 vs. entorhinal). To preview, this analysis indeed revealed a 3-way interaction of Hemisphere by Changes by ROI (Results). We thus tested our hypotheses regarding a shift in CA1 connectivity in each hemisphere separately.” (p. 28). We further changed the order of reporting, so that we report this ANOVA before focusing on the left hemisphere (pp. 8-9). We note that we also found our effect in the left hemisphere, consistent with the previous studies that did report lateralization.

7. For the data shown in Figure 2 (line 198 in text), an ANOVA is reported which is presumably the main effect of # of changes, and this is described as testing whether there was a significant increase or decrease in connectivity with the number of changes, but the ANOVA just shows that connectivity varies across conditions (which is a useful

point, but slightly different from how the authors frame the result). The mixed effects model described later does test the linear trend, so the ANOVA should just be described more neutrally.

Thank you. We now describe the different directions of change qualitatively, and the ANOVA more naturally: “As predicted, CA1-entorhinal connectivity increased with more changes (Figure 2). By contrast, and again consistent with our predictions, CA1-CA3 connectivity decreased as number of changes increased (Figure 2). One-way repeated-measures ANOVAs with the factor of Changes (0-4) conducted for each pair of regions separately, confirmed that in both region pairs (CA1-entorhinal or CA1-CA3), Changes significantly modulated connectivity (CA1-entorhinal: $F_{(4,72)} = 4.49, p < .003, \eta_p^2 = 0.20$; CA1-CA3: $F_{(4,72)} = 3.58, p < .02, \eta_p^2 = 0.17$)” (p. 9).

Reviewer #2 (Remarks to the Author):

The manuscript seeks to understand how pattern completion/separation might relate to unexpected feedback during retrieval of recently learned room layouts or objects within rooms. The authors sought to test a model developed in rodents that suggests that ERC-CA1 interactions might be important to encoding (pattern separation) and CA3-CA1 connectivity important to retrieval (pattern completion). The authors sought to test this model by having subjects learn rooms, which they then retrieved in the scanner. The subjects received feedback that either matched or mismatched the cue by 1-4 features. The authors found that (left) ERC-CA1 connectivity increased with mismatches and (left) CA3-CA1 connectivity decreased with mismatches. The correlation between the putative reinstated trace and the actual room was not different from the correlation for mismatched rooms. CA1 multivariate pattern differences for each subject as a function of room correlated with ERC-CA1 connectivity although there was no negative correlation for CA3-CA1. There was also a tendency of mismatched features to be lower for CA1 multivariate patterns than the same room. The authors conclude that gating by different paths within the hippocampus (ERC-CA1 vs. CA3-CA1) may be important for mediating pattern completion and pattern separation dynamically during tasks.

The patterns of findings are interesting here and testing the models presented, even if their exact mapping in the task is slightly unclear, is potentially useful. There are some concerns, however, with the results and statistical differences/consistency with past work on this topic. The theoretical motivations also appear somewhat unclear in places. The MRI methods may also be suboptimal in some cases. Aspects may be of interest to a broad audience, although other aspects seemed better suited for an audience with experience in memory. I detail some of these concerns below.

MAJOR

1) Given some of the past work on the topic, it seems important to replicate that basic finding that univariate findings differ as a function of match vs. mismatch (Duncan, Ketz, Inati, & Davachi, 2012). Did this effect replicate?

We thank the reviewer for this comment and regret not having made this clearer in the original paper. This paper is a re-analysis of the previously reported data by Duncan et al. (2012). We now reference those results in the manuscript again and indeed, we reproduced these results. Specifically, we conducted a linear trend analysis on the average univariate activation in CA1 in each level of changes, and obtained a significant linear trend ($t_{(18)} = 2.43, p < .05$). We now report this result in the Results section (p. 10).

If so, how did (or did not) univariate pattern relate to multivariate patterns reported here?

It is not entirely clear what specific elements of our results are being queried here. For our main results, however, and in response to a similar question by Reviewer #1, we now have directly assessed whether our connectivity and multivariate findings still explain variance when we statistically control for univariate activation. These are described below:

Regarding the relationship between univariate activation and our main connectivity findings, we note that the univariate activation in CA3 or in entorhinal cortex was not modulated by the level of changes. Thus, univariate activity cannot account for our connectivity findings. And, since CA1 connectivity with these regions was modulated in opposite directions, CA1 activation also cannot account for our connectivity findings. Nevertheless, to respond more directly to the concern about univariate activation and to establish that the connectivity findings are independent of univariate activation findings, we have also now run an additional control analysis: for each pair of regions (CA1-EC/CA1-CA3) we adopted a mixed-effects linear model with connectivity in each level of changes as the explained variable, and as explaining variables we included the average univariate activation per participant in each of the regions in the pair (i.e., CA1 and EC or CA1 and CA3), as well as the interaction of univariate activation. In addition to the univariate activation, as our main contrast of interest we included in the model as an explaining variable either the linear trend contrast (0-4 level of changes were coded as -2,1,0,1,2) or the match < mismatch contrast (0-change was coded as -4, whereas 1-4 level of changes coded as 1). As a reminder, we found that for CA1-EC connectivity, the match < mismatch contrast

better accounted for the pattern of connectivity changes thus we included the match < mismatch contrast as our contrast of interest. For CA1-CA3, a linear contrast better accounted for the connectivity changes, thus we included the linear trend contrast as our contrast of interest. Including univariate activation in our models together with these contrasts (i.e., match < mismatch/linear trend) allowed us to test whether the differences in connectivity across levels of changes are significant also when statistically accounting for univariate activation. In these models, the results remain consistent. Specifically, we compared these full models including both our contrast of interest and univariate activity to models that only included univariate activity (additionally, an intercept per participant was included in all models). We found that for both pairs of regions, the models with the contrast of interest significantly explained more variance compared to the models including only univariate activity (CA1-EC: a model including the match < mismatch contrast: $\chi^2 = 14.32$, $p < .001$, AIC or BIC reductions > 10; CA1-CA3: a model including the linear contrast: $\chi^2 = 11.26$, $p < .001$, AIC or BIC reductions > 6). These results show that for both CA1-EC and CA1-CA3 connectivity, the levels of changes significantly explain variance when controlling for univariate activation. Thus, they suggest that our main connectivity findings are unlikely to be explained by univariate activity in any of the three regions involved. We now report this control analysis in the Results (pp. 10-11).

For the correlation of CA1 reinstatement with CA1-entorhinal connectivity, we now additionally control for univariate activation, both during the cue and during the image. Specifically, we conducted a multiple linear regression that included the match < mismatch connectivity contrast as the explained variable, and as explaining variables we included the CA1 reinstatement measure, univariate activation in CA1 during the cue, and during the match images, as well as the match < mismatch univariate contrast in CA1 and in EC (to account for univariate activation for connectivity). Even when including all of these other factors, the reinstatement variable was significant ($t_{(13)} = 1.93$, $p < .05$). Thus, the correlation is unlikely explained by univariate activation differences between participants. This analysis is reported in the Supplementary Information.

We also conducted two control analyses for the prediction error analysis, which is now reported in the Supplementary Information: First, in line with reviewer #1's suggestion, for each cue, we subtracted from the correlation of the cue and its corresponding image, the average correlation of that same cue to all other images that are in the same type of trial (i.e., same task, and levels of changes), but correspond to other images. Any activation level differences between levels

of changes should appear in the other images of the same level of changes, and thus cancel out by the subtraction. Importantly, like in our main analysis, the match > mismatch contrast was significant ($t_{(18)} = 2.60, p = .018$). Second, to further control for univariate activation, we conducted a multiple regression model in which the prediction error match > mismatch contrast per participant was the explained variable, and as explaining variables we took the same match > mismatch contrasts, but computed for the cue activation and the image activation in the left CA1. Note that here, the intercept reflects the effect of interest, namely, whether the match > mismatch difference is significantly different from 0, when controlling for univariate activation. Indeed, the intercept was highly significant ($t_{(16)} = 3.46, p < .004$), suggesting that CA1 multivariate prediction error signal cannot be explained by univariate activation.

It is also notable that the hippocampal multivariate effect (Figure 4) does show decreases in pattern similarity but not in the linear fashion reported in some studies (Lacy, Yassa, Stark, Muftuler, & Stark, 2010; Stokes, Kyle, & Ekstrom, 2015).

We thank the reviewer for the comment. However, we think there is a misunderstanding of the data reported in those papers and want to clarify. Lacy et al. (2010) report univariate activity, and not a multivariate data. Further, we note that, in our read of their paper, the effect was not linear. Lacy et al. (2010) show that in CA1, similar lures lead to repetition suppression to the same extent as identical repetitions. In our view, linearity cannot be addressed in the previous study, as there are only two levels of change (identical or similar lures). From the data reported – it seems like CA1 does show suppression for similar lures and identical images, thus a non-linear effect. This univariate effect may suggest that CA1 does not differentiate between identical repetition and lures, while we show that it does (namely, our prediction error analysis showing that similarity between match images is higher compared to altered versions of that image). To us, this may suggest that the univariate method used by Lacy et al. (2010) may not be sensitive enough, and that more refined tools (RSA) may be better suited to detect differences in CA1 responses to change. Another possibility is that the difference in the paradigm used underlies the lack of an effect in Lacy et al.'s results. One interesting difference is that our paradigm and others (Kumaran et al., 2006, 2007; Chen et al., 2011, 2015) that have detected univariate differences involved a prediction that is then violated – compared to no such prediction in Lacy et al.'s designs. This difference between the paradigms may further strengthen the argument for a role of CA1 in detecting mnemonic prediction errors.

Stokes et al. indeed did use RSA to measure the representation of a different spatial environments (cities in a square, circle, or morphed between a circle and a square shape). In most of their analyses, however, they do not report reduced pattern similarity in CA1, aside from one case. CA1 showed reduced similarity between morphed cities, compared to between “regular” shapes, namely, circle and square cities. First, since RSA was computed as participants were viewing these cities, it is unclear to what extent these differences reflect memory reinstatement rather than perception, making the two studies very different. More importantly, this result simply states a difference between the two conditions (regular vs. morphed shapes), thus, it is unclear how linearity could be inferred from that difference? Furthermore, another significant difference between our paradigm and this one is that we are looking at similarity between a verbal cue (e.g. Johnson’s boys bedroom) and viewing of the actual room, while this study examined similarity between two images. To conclude, we believe that the differences in approaches and paradigms between these studies and our study make it rather reasonable that our results are different, and further emphasize the novelty of our findings.

Even here, the match-mismatch effect is relatively weak ($p < .04$). A searchlight analysis could potentially help better resolve if there is a signal to noise issue here with the CA1 ROI or potentially something also present in CA3 (see point 2).

We thank the reviewer for suggesting that we might benefit from an approach that allows for better SNR. We note that a searchlight might not be advised in our case, since a searchlight may not respect the boundaries of oddly shaped subregions of the hippocampus. Further, and more critically, searchlight would require registration of the ROIs to a template for group analysis, which not only distorts the data (and adds the concern that subfields might bleed to one another), but also requires some smoothing, that may occlude RSA effects (Kriegeskorte, 2008). Thus, we feel that an ROI approach to RSA analyses is better advised.

However, we did follow the reviewer’s advice, and to increase our SNR in our RSA analyses, we excluded voxels that demonstrated low responsiveness to our task. Specifically, for each participant, we ran an independent model that estimated task activation, in each level of changes. We then averaged on levels of changes to get the average activation level per voxel in our task. Then, for each participant, we excluded the bottom third of voxels in region CA1 from our RSA analyses. This type of voxel selection has been previously used in multivariate studies (Favila et al., 2016; Chanales et al., 2017). Critically, if our results were

obtained by chance, selecting voxels should not have changed our result, or even make it worse since we exclude some of the data. If, however, our results reflect a true effect masked by potentially noisy/non-responsive voxels, then selecting voxels should boost SNR, and thus improve our ability to detect an effect. Indeed, we have obtained a highly significant prediction error effect. The match > mismatch contrast was significant at $p = .0056$. We now report this analysis in the Results (p. 16). For completeness, we additionally report the same analysis without voxel selection in the Supplementary Information. As noted above, this analysis further held when controlling for univariate activation or when subtracting the pattern of other room images, further demonstrating the robustness of this result. These control analyses are reported in the Supplementary Information.

While ERC-CA1 connectivity showed a positive trend and CA3-CA1 showed a negative trend, were these slopes different?

Yes, we did report an interaction of region (CA3-CA1/ERC-CA1) by number of changes, which qualifies that the effect of changes differed between the two region-pairs. We now ran an additional analysis in which we computed the linear contrast per participant per regions-pair (CA3-CA1/ERC-CA1), as well as the match vs. mismatch contrast (i.e., the slope of the difference in connectivity per subject, as characterized by these two contrasts). We computed both of these slopes because the match/mismatch slope explained more variance in the ERC-CA1 region-pair, while the linear contrast explained more variance in the CA3-CA1 region-pair, thus we wanted to account for both. We then directly compare these contrasts between region-pairs using paired-sample t-tests. Indeed, for both types of contrasts, the slopes were significantly different (p 's < .002). Given that our interaction analysis provides a straight-forward demonstration of the same idea, we currently do not include the analyses detailed here in the manuscript, but we are happy to include it if the reviewer recommends it.

It is also notable that CA3-CA1 connectivity did not correlate with CA1 multivariate pattern differences (unlike ERC-CA1). This might have been expected if CA3-CA1 connectivity was mediating the effect but was not found, suggesting that ERC-CA1 (and not CA3-CA1) input matters to CA1 neural patterns.

More generally, given that the authors did not find CA3-CA1 connectivity correlations with CA1 multivariate patterns, it seems important to try to better understand why not. It is also notable that the correlation with the reinstated room and the actual room image did not differ significantly (line 247). This is somewhat of a concern because if CA1 contains information about the trace and its mismatch, this effect should likely have been there. The authors attribute this to variance but other studies have found such effects, at least in the hippocampus (Mack & Preston, 2016). Thus, it was surprising not

to see this in the data in some form. Finally, the authors use one-tailed t-tests in many places. These should be avoided as they have a high risk of false positives.

We thank the reviewer for the comments. Regarding the first point that we do not see a significant same-room vs. other rooms difference, following the reviewer's advice and with a closer look at the data enabled by voxel selection (same voxel selection as was detailed above), we ran the same reinstatement analysis comparing the similarity of the cue to the corresponding match room versus the average similarity of the cue to match rooms that corresponded to other rooms. As now reported in the Results (p. 13), we do indeed see a significant reinstatement effect ($t(18) = 2.5$, $p = .01$, one-tailed; note that the result is significant with a two-tailed test as well, $p = .02$). This result suggests that the lack of an effect previously was at least partly due to the inclusion of weakly active perhaps noisy, voxels. We want to thank the reviewer for their suggestions here.

We also note that previous studies (Mack et al., 2016; Tomparry et al., 2016) that have reported significant reinstatement used images as cues during retrieval. In our design, we measure memory reinstatement when participants are viewing a verbal cue, and we computed the similarity of this verbal cue to highly complex and detailed stimulus of a room, including a specific layout and multiple furniture. Nevertheless, reinstatement is observed, demonstrating the robustness of our findings.

The reviewer further mentioned the lack of a relationship between reinstatement and CA1-CA3 functional connectivity. We agree it is important to understand why CA1-CA3 did not correlate with reinstatement. While we want to avoid too much speculation about a null effect, we have offered a possible interpretation in the Discussion by which CA1 output to VTA may help to synchronize CA1 processing with entorhinal cortex, but not with CA3, suggesting a closer link between CA1 reinstatement and CA1-entorhinal connectivity, compared to CA1-CA3 connectivity. We now specifically mention this in the paragraph discussing our interpretation: "This interpretation of synchronization via VTA input to CA1 and entorhinal cortex is further consistent with the lack of a correlation between CA1-CA3 connectivity and CA1 prediction strength". (p. 20).

As we detail above, one-tailed t-tests are justified when there are a priori directional hypotheses, rather than testing for any difference that allows the rejection of the null-hypothesis, as is the case in two-tailed tests (Ruxton and Nauhauser, 2010). Of course, these a priori reasons need to be detailed so that

they can be evaluated by readers. We have outlined those priors above (in response to Reviewer #1), and we repeat them here:

Specifically, for the reinstatement measure:

- 1. Prior research has shown memory reinstatement in CA1 (Mack et al., 2016; Tompary et al., 2016).**
- 2. We computed the difference between similarity of the cue to the intact image of the same room (e.g., the cue of Johnson's boy bedroom with the intact image of Johnson's boy bedroom) to similarity of the cue to other rooms (e.g., the cue of Johnson's boy bedroom with the intact image of the Smith's living room). Clearly, the cue should be more similar to the intact image of the same room compared to others, and there is no reason to believe the other direction (more similarity to other images) would occur.**

In any case, we further note that the reinstatement measure is also significant using a two-tailed t-test.

Regarding the correlation measure:

The computational models that motivated our work (Hasselmo, 2002; Meeter 2004) provide strong theoretical foundations for a directional hypothesis.

Specifically, they argue that more novelty - and specifically, stronger mnemonic prediction errors, should lead to a shift towards an encoding 'state' mediated by CA1-entorhinal connectivity. Thus, we hypothesized that prediction errors should specifically increase CA1-ERC connectivity while also decreasing CA1-CA3 connectivity - both are directional hypotheses and warrant one-tailed tests.

We agree with the reviewer that our reasoning might have not been laid out clearly enough in the manuscript and thus made it harder to evaluate that our one-tailed t-tests are justified. We now further motivate the one-tailed tests and specify the reasons in the Methods section (p. 30), and thank the reviewer for raising this concern.

2) The authors reported using standard MPRAGE anatomical sequences (line 453) and then co-registering a high-resolution EPI to this sequence. However, there are reasons one might expect the MPRAGE to give insufficient signal to noise and contrast to identify the subfields. For one, T2 TSE are considered the gold standard for identifying subfields and an MPRAGE is unlikely to have the differences in contrast/signal needed to discriminate CA1 from CA3/DG on most slices (Wisse et al., 2016). The concern is that at least some of the weak/non-significant effects could be driven by poor subfield identification. The authors should consider a searchlight analysis. Ideally, a T2 image would be available for more precise segmentations.

Unfortunately, T2 images were not collected. As detailed above, searchlight might not be advisable for subfields. While we acknowledge the reviewer's concern, our voxel-selection approach did prove useful in potentially eliminating noisy voxels. Critically, it is important to note that, in our connectivity findings we see strong opposite effects using these regions. If poor spatial identification of these regions was a major issue, it should have impeded our ability to see such effects.

3) Theoretical: The authors base the ideas of the study on a model developed and tested largely (but not exclusively) in rodents. However, one issue is that the single neuron/LFP signals studies by groups like the Colgin lab don't have a clear mapping onto the BOLD signal, which is likely a mix of input from different subfields (Logothetis, 2003). While testing the model is reasonable, it is not clear that what the authors are observing with connectivity is an actual up-weighting of one pathway and down-weighting of the other. This is accentuated by the fact that CA3-CA1 connectivity did not correlate with CA1 multivariate patterns. As such, it is difficult to link the findings precisely to the model mentioned in the introduction.

We think it is important for any animal model of the human brain to be translated and tested in the actual human brain. Our previous work (Duncan et al., 2014) has shown that connectivity between CA1 and CA3 is significantly higher during retrieval compared to novel encoding blocks. Furthermore, something often difficult to do in animal studies is to obtain good behavior to correlate with the brain data. In that prior paper, we linked BOLD connectivity in distinct CA1 pathways with encoding and retrieval success as measured behaviorally. So, even though our measure is an indirect and inferred measure of underlying neural activity, that prior study lays the groundwork for our experiment here and provides strong support for our interpretation. Namely, our prior work shows that differential CA1 connectivity is associated with encoding and retrieval operations.

Here we extend that result to see if CA1 connectivity dynamically changes with the nature of the sensory evidence provided in response to retrieved memory representations. Of course, we make no claims in the paper about single or multi-unit results as we are measuring BOLD signal. However, the Logothetis paper mentioned by the reviewer does show a nice correlation between BOLD and LFPs. We agree that this point makes it possible/likely that the BOLD signal coming out of one region is a mix of activity from that region + inputs to that region. However, the connectivity measures we adopt should be less susceptible to mixing effects since we are looking for correlated activity across regions. Also, if we are measuring a 'mix' we may not have expected to get the results we do get which show connectivity effects in opposite directions across CA1-CA3 versus CA1-EC, and are entirely consistent with these theoretical models.

However, in response to the reviewer's comment, we conducted an additional analysis aiming to statistically dissociate the monosynaptic pathway from the trisynaptic pathway in the hippocampus. The monosynaptic pathway refers to a direct synapse between ERC and CA1. The trisynaptic pathway is the pathway leading from ERC through DG (dentate gyrus) and CA3 to CA1. Naturally, we cannot directly measure synaptic activity. Critically, however, we can examine whether CA1-ERC correlated activity might correspond more to the monosynaptic pathway, by showing that this connectivity cannot be explained by connectivity patterns in subregions along the trisynaptic pathway. With that rational in mind, we conducted a mixed-level multiple regression analysis in which CA1-ERC connectivity was the explained variable. As explaining variables, we included the match < mismatch contrast, in addition to ERC-CA3 (CA3 and DG are collapsed together) connectivity, and CA3-CA1 connectivity, which compose connectivity along the trisynaptic pathway. We then compared this model to a model including only the other connectivity values, but without the match < mismatch contrast. We found that the full model including the match < mismatch contrast significantly outperformed the model including only connectivity between the other subregions (AIC difference of 11, BIC difference: 9, $\chi^2 = 13.05$, $p < .0005$; replacing the match < mismatch contrast by a linear contrast yielded similar results). This result shows that the modulation of CA1-ERC connectivity by levels of changes is statistically independent from connectivity of nearby subregions. It suggests that the increase in CA1-ERC connectivity in response to changes might reflect the monosynaptic pathway, more so than the trisynaptic pathway. Aside from demonstrating the robustness of our findings, this analysis lends further support for the dissociation between connectivity in different hippocampal subfields.

Finally, while CA1-CA3 connectivity indeed did not correlate with our measure of CA1 reinstatement, it is, importantly, modulated by prediction error as experimentally manipulated. The strongest test of the theoretical models was by using the task in which we manipulated changes experimentally. Using our experimental manipulation of levels of changes, we found an increase in CA1-ERC connectivity, versus a decrease in CA1-CA3 connectivity. These data are consistent with the models, and thus provide supporting evidence that mnemonic prediction errors shift hippocampal states. The reasons for why CA1-CA3 connectivity did not correlate with our measure of CA1 reinstatement are currently unknown. We have some speculations, as detailed above, and in the Discussion (p. 20).

In addition, pattern completion and separation are almost certainly happening during both encoding and retrieval. So for example, retrieval is likely to involve both separation (separating the memory from the other interfering/competing ones) and completion to the desired output. This is likely to happen very rapidly and it seems unclear whether BOLD can completely capture these given the relatively poor temporal resolution. Of note, the design nicely attempts to separate these aspects, however, it is not clear the results can directly contact this aspect of the Hasselmo/Colgin models.

We are in complete agreement with the reviewer that pattern completion and separation might co-occur during both encoding and retrieval. Critically, we are upfront in the paper that what we are measuring is differential connectivity triggered by matches versus mismatches. We do not and cannot link these connectivity findings with the processes of pattern separation and completion. As implied by the reviewer, we agree that many different computations may support encoding and retrieval. The specific computation that is performed by this inter-region communication is a fascinating topic for future research, likely in other methods and experimental designs. While we do mention pattern separation and completion in the Introduction as computations that are canonically associated with encoding vs. retrieval, we do not argue in the manuscript that these are the specific computations performed by the connectivity we observed.

MINOR

1) The rationale for the findings being in left hemisphere was a little unclear. The authors should be sure to test full models with hemisphere as a factor first and then conduct post-hoc analyses accordingly.

There was no a priori rationale for the findings to be lateralized to the left hemisphere but since many processes supported by the hippocampus in human work tends to be lateralized, we always take this into account when conducting our data analyses. As a short review, previous univariate mismatch findings do not show consistent lateralization. Specifically, while Dudukovich et al. (2011), as well as Kumaran et al. (2006,2007) found a prediction-error effect in the left CA1, Chen et al (2011, 2015) collapsed across hemispheres and used one large ROI so it is hard to know if the effects were driven by one hemisphere.

Thus, in the original paper, we did include hemisphere in an ANOVA testing for laterality effects. Functional connectivity was entered to a repeated-measures ANOVA with Level of changes (0-4), ROI (CA1-ERC/CA1-CA3), and Hemisphere (left/right). Indeed, this ANOVA revealed a three-way interaction of Level of changes by ROI by Hemisphere, indicating the differential influence of Level of

changes on ROI varied between the two hemispheres. This interaction qualifies our approach of analyzing each hemisphere separately when testing for our main hypotheses regarding a decrease in CA1-CA3 connectivity, and an increase in CA1-ERC connectivity. While this ANOVA was reported in the manuscript previously, we acknowledge that our rationale was not laid out clearly. We now explain this rationale in the Methods: “Since previous univariate findings do not show consistent lateralization (Chen et al. 2011, 2015; Dudukovich et al. 2011; Kumaran et al. 2006,2007; see also Jordan, 2019), we reasoned to look at each hemisphere separately. Thus, we first conducted an ANOVA of Hemisphere (left/right) by Changes (0-4) by ROI (CA3 vs. entorhinal). To preview, this analysis indeed revealed a 3-way interaction of Hemisphere by Changes by ROI (Results). We thus tested our hypotheses regarding a shift in CA1 connectivity in each hemisphere separately.” (p. 28). We further changed the order of reporting, so that we report this ANOVA before focusing on the left hemisphere (pp. 8-9). We note that we also found our effect in the left hemisphere, consistent with the previous studies that did report lateralization.

In addition, one might have expected right sided activation given the spatial nature of the retrieval component, although such lateralities are inconsistent with fMRI.

In line with the reviewer’s comment on inconsistencies in lateralities, and as detailed above, previous univariate mismatch findings do not show consistent lateralization. Specifically, while Dudukovic et al. (2011), as well as Kumaran et al. (2006,2007) found a prediction-error effect in the left CA1, Chen et al (2011, 2015) reported one bilateral ROI. Of note, these previous studies reporting a lateralized effect included object stimuli and either an orthogonal n-back task (Kumaran et al., 2006, 2007) or a paired-associate match detection (Dudukovic et al., 2011). Thus, the retrieval component in these paradigms was potentially less spatial compared to our task. Nonetheless, we observed a left hemisphere laterality, consistent with these previous studies. We further note that we did not find a difference in connectivity between the Layout task, which is more spatial in nature, and the Furniture task, which is item-focused (Results, p. 8). Together, these results suggest that mismatch signals in the hippocampus might be stronger in the left hemisphere, and potentially independent of tasks. Future work is needed to better understand the nature and consistency of hippocampal laterality in function.

2) ““In CA1-entorhinal connectivity, we found that the full model significantly outperformed the linear model ($\chi^2 = 4.39$, $p < .05$), but not the match < mismatch model ($\chi^2 = 1.31$, $p > .25$), suggesting that the match < mismatch contrast better describes CA1-entorhinal connectivity.”

Ideally, AIC/BIC scores could be used here to deal with potential differences in degrees

of freedom, unless these are identical?

The full model included both the linear trend contrast (0-4 level of changes were coded as -2,1,0,1,2) and match < mismatch contrast (0-change was coded as -4, whereas 1-4 level of changes coded as 1) as explaining variables, in comparison to the linear model that only included a linear contrast (in both, functional connectivity was the explained variable). Thus, the full model had one additional degree of freedom, warranting the reporting of AIC or BIC scores. Note that the use of AIC vs. BIC scores is contentious, and it is intensely debated which one is more useful in model selection. Here, we adopted AIC, following the suggestion that AIC may provide a better test under the assumption that the real model in the world is unknown, and is not a part of the candidate models (compared to BIC, which assumes that one of the models being compared is the real model in the world, an assumption that we and others find unreasonable). However, for completeness, we report both (Results, p. 10). In CA1-Entorhinal, AIC was lower for the full model compared to the linear model (full: -80.68, linear: -78.29; BIC scores were comparable: full: -67.91, linear: -68.07), suggesting that the match < mismatch contrast better describes CA1-entorhinal connectivity than the linear contrast. We note that we used the same model comparison procedure in CA1-CA3 connectivity, thus we now also report AIC scores for CA1-CA3 connectivity. Note that in CA1-CA3, we found that the full model better accounted for connectivity compared to the match < mismatch contrast, but not compared to the linear contrast, suggesting that in CA1-CA3, the linear contrast better thus we now also report AIC scores. Accordingly, AIC was lower for the full model compared to the match < mismatch model (full: -72.56, match < mismatch: -65.93; BIC: full: -59.80, match < mismatch: -55.71; Results, p. 10).

References

- Duncan, K., Ketz, N., Inati, S. J., & Davachi, L. (2012). Evidence for Area CA1 as a Match/Mismatch Detector: A High-Resolution fMRI Study of the Human Hippocampus. *Hippocampus*.
- Lacy, J. W., Yassa, M. A., Stark, S. M., Muftuler, L. T., & Stark, C. E. (2010). Distinct pattern separation related transfer functions in human CA3/dentate and CA1 revealed using high-resolution fMRI and variable mnemonic similarity. *Learn Mem*, 18(1), 15-18.
- Logothetis, N. K. (2003). The underpinnings of the BOLD functional magnetic resonance imaging signal. *J Neurosci*, 23(10), 3963-3971.
- Mack, M. L., & Preston, A. R. (2016). Decisions about the past are guided by reinstatement of specific memories in the hippocampus and perirhinal cortex. *Neuroimage*, 127, 144-157.

Stokes, J., Kyle, C., & Ekstrom, A. D. (2015). Complementary roles of human hippocampal subfields in differentiation and integration of spatial context. *J Cogn Neurosci*, 27(3), 546-559.

Wisse, L. E., Daugherty, A. M., Olsen, R. K., Berron, D., Carr, V. A., Stark, C. E., et al. (2016). A harmonized segmentation protocol for hippocampal and parahippocampal LID - 10.1002/hipo.22671 [doi]. *Hippocampus*.

Reviewer #3 (Remarks to the Author):

The authors present work demonstrating differential functional connectivity between CA1 and CA3/DG and Entorhinal cortex during a scene change detection task, and evidence that this reflects a change in the network state dependent on prediction errors for the scenes.

Overall, I greatly enjoyed reading this manuscript. It was clear and well-written, and the motivation for the work is well founded. Analyses were generally laid out in a very principled manner, with one typically addressing alternative interpretations left over from the preceding analysis.

The results are original and of potentially high impact.

I found myself with only one category of related comments and concerns, which should be straightforward to address:

1) There are several allusions to parametric relationships with room similarity in the manuscript, usually presented as a follow-up test on a significant ANOVA effect of # of changes. It would seem the range of these tests was guided by the authors' predictions (i.e., change/no change vs linear), which is great to see and easy to follow – but the ANOVA and visual inspection of the data leave open the possibility that the relationships actually take other untested forms.

Specifically, when scrutinizing the behavioral data, one could walk away with two impressions about the change manipulation: from an accuracy perspective, it appears change levels 1-3 are fundamentally different from 0 and 4. From the RT data it seems as though participants' default strategies are to assume that there is a change in the scene and they search until they rule this out or confirm – and interestingly, only with 3-4 changes does the perceptual mismatch map onto a change in speed. Both of these suggest a more step-wise influence of the experimental manipulation (although at somewhat different change levels for ACC vs RT). Could it be the case that participants move from a serial search to a more holistic scene matching process at different levels of change? Likewise, it's a bit tricky to think about when mismatches are actually going to be well-represented in the neural data (i.e., is there a sharp boundary at 2-3 or 3-4?).

a. It would be helpful to know the simple effects here between levels (and in the subsequent neural analyses across change levels)

We thank the reviewer for the positive evaluation of our work and for the interesting insightful observations about the behavioral data. Following the reviewer's advice, we examined the simple effects between level of changes. In accuracy rates, the 0-changes and 4-changes were significantly different from 1-3 changes (p 's $< .0001$). The 1-change condition further differed from 2 and 3-changes ($p < .05$, $p < .002$, correspondingly). Other comparisons were not significant (p 's $> .15$). When looking at reaction times, we found that reaction times in the 0-changes condition were shorter than 1-change ($p < .05$), but longer than 3-changes ($p < .01$), and marginally longer than 4-changes ($p < .08$). RTs in the 1- and 2-changes conditions were significantly slower than both 3- and 4-changes (p 's $< .001$). There was no difference between 1- vs. 2-changes, or between the 3 and the 4-changes conditions (p 's $> .2$). Thus, the pattern of results agrees with the observation of the reviewer: indeed, and as we have mentioned in the manuscript (p. 7), in accuracy, it seems like the 0 and 4 levels of changes are very different from 1-3, probably because it was easier to detect matches or mismatches at these extremes. Regarding RTs, the reviewer's intuition that participants adopt a strategy to assume a change until they can rule it out indeed seems reasonable at first glance. However, as we understand it, this intuition would predict that the 0-changes condition would take the longest to respond, as participants would have to make sure that there were no changes. In contrast, RTs in the 0-changes condition were significantly shorter than the 1-changes condition, suggesting that participants might have not adopted such a strategy. But we also appreciate the reviewer's suggestion that participants may be switching between serial search strategies to a more holistic strategy.

Regarding the neural data, the pattern of simple-effects interestingly did not correspond with the simple effects in behavior. Specifically, in CA1-CA3, in addition to the linear trend, the difference between 0-changes and 4-changes was highly significant ($p < .001$) with the 4-changes condition also being significantly different from 1 and 2 changes ($p < .05$) [and marginally different from 3-changes condition ($p < .1$)]. The 3-changes condition was marginally different from 1-change ($p < .1$). Other comparisons were not significant (p 's $> .15$). In CA1-ERC, all levels of changes were significantly different from 0-changes (p 's $< .01$), with no difference between them (p 's $> .3$). We thus note that in both regions-pairs, the pattern of the neural data was different from either the accuracy or the RTs pattern. For accuracy, this would have predicted a difference between 0-changes and 1-3 changes, and between 4 changes and 1-3 changes, but with no difference between 0 and 4 changes, which is not the data we observed in either of the regions-pairs. Regarding the pattern of RT results, this pattern would have

predicted no difference within the levels of 0-2 changes or within the levels 3-4 changes, but with a difference between 0-2 changes versus 3-4 changes. Again, the pairwise comparisons in both regions were very different. Thus, while it is definitely possible that participants might have adopted some different strategies at different levels of changes, accuracy or RTs are unlikely to explain our connectivity findings.

To further address this possibility, we conducted a mixed-level models that included accuracy and RTs, and we are happy to report that our functional connectivity results hold. Specifically, for each pair of regions (CA1-ERC/CA1-CA3), we included connectivity as the explained variable, and accuracy rates and RTs per participant and per each level of changes as explaining variables. In line with the contrasts that explained more variance in connectivity, in CA1-ERC, we added the match < mismatch contrast as our primary explaining variable, and in CA1-CA3, we added the linear contrast our primary explaining variable. We then compared these full models to models that included only the behavioral factors as explaining variables. Our model comparisons showed that for both pairs of regions, the full model that included the primary contrast of interest significantly outperformed the models that only included the behavioral data, without the contrast of interest (CA1-ERC: $\chi^2 = 10.29$, $p < .002$, AIC or BIC reductions > 5; CA1-CA3: $\chi^2 = 12.18$, $p < .001$, AIC or BIC reductions > 7). We further note that comparing the full model to a model including only the match < mismatch/linear trend contrasts did not significantly explain variance, suggesting that accuracy and RT do not explain variance in connectivity beyond the variance explained by level of changes (CA1-ERC: $\chi^2 < 0.03$, $p > .9$, AIC or BIC were lower for the simpler match < mismatch model that did not include the behavioral factors, differences > 3; CA1-CA3: ($\chi^2 < 1.13$, $p > .56$, AIC or BIC were lower for the simpler model, including only the linear model).

We now fully report these additional analyses in the Supplementary Information, and mention them in the manuscript after reporting the connectivity results: “We further statistically controlled for accuracy and reaction times differences between participants and levels of changes, and our results held (Supplementary Information).” (p. 11). Together, these analyses show that accuracy rates or reaction times are unlikely to account for the connectivity findings.

b. Would the authors expect a more clear mismatch signal for their neural data if only examining the subset of trials for 1-4 where participants were correct (indicating they did ultimately detect the change)? If this was a flat distribution, and different from the one reported in which the hits and misses are combined in the data (if I understood the methods correctly), this would lend further support to the gradient in the reported data being driven by the proportion of trials where the mismatch was detected.

We thank the reviewer for this comment. We note that in the current study, we were agnostic to the question of how connectivity changes might interact specifically with participants' match/mismatch judgements. Accordingly, and as reported in the paper, connectivity was calculated based on all trials, regardless of accuracy. However, in response to the reviewer's question, we have now calculated connectivity including only correct trials. Below are the functional connectivity results, calculated based on only accurate trials, with the original graphs including all items (correct and incorrect) presented next to them. Note that in both pair-regions, taking only accurate trials (right side) resulted in a similar pattern of connectivity to including all trials in the analysis (left side).

To further examine the reviewer comment, we compared connectivity for accurate versus inaccurate trials. Note that this was not the original purpose of our investigation; in fact, we designed the study so that our participants will have very strong memories of the rooms, and overall high accuracy. Thus, we had low number of inaccurate trials in the 0- and 4-changes conditions, which prevented us from examining connectivity for inaccurate trials in these levels of changes. We did, however, compare connectivity for inaccurate trials in 1-3 changes. As can be seen below, within each level of 1-3 changes, there was no difference between accurate vs. inaccurate trials. Together with the data above that statistically controlled for accuracy and RTs, these data suggest that our

connectivity findings likely reflect a broad match/mismatch signal, and are robust to participants' decisions. Since we did not find a difference between accurate and inaccurate trials (where we could test for such differences in the 1-3 changes), we did not change our original approach in the manuscript, and collapsed included all trials in our analyses to increase power. As our study was not specifically designed to address the interaction of connectivity with accuracy in the match-mismatch detection task (which would necessitate a comparison between accurate and inaccurate responses, and enough trials in each response-type per level of changes), we prefer not to include these additional analysis in our manuscript. We nevertheless agree that the question of how and match/mismatch signals interact with behavioral measures of mismatch detection is a compelling topic for further research.

2) Largely the same comment, but applied to the functional data, I found it interesting that the CA1-CA3/DG pattern – although tested as linear or match-mismatch – really appears to exhibit a gradient starting at 2-3 changes (not unlike the behavioral steps occurring in that range). Here it would again be nice to know the simple effects coming out of the significant ANOVAs, and I began to wonder if an alternative “change/no change” contrast motivated by the behavior rather than the design would better explain the relationship between CA1-CA3/DG. There has been work mapping decision thresholds and memory judgment confidence to different components of the declarative memory system in episodic memory tasks (e.g. <https://www.ncbi.nlm.nih.gov/pubmed/23019246>, with some evidence linking these concepts to MTL structures as well when searching semantically-familiar scenes for personally-relevant episodic cues - <https://www.nature.com/articles/s41598-018-24549-y>) and I wonder if such a behaviorally-grounded analysis strategy, combined with the authors' between-subjects correlation analysis, could give more leverage over the response profiles across change level.

We thank the reviewer for suggesting these additional analyses. We addressed the simple effects in connectivity, as well as using behavioral measures as predictors of functional connectivity in our response above. Briefly, we did not find evidence that the differences between levels of change in CA1-CA3 corresponded to the accuracy or the RT data. Thus, while some components of the episodic memory network surely correspond strongly to confidence judgements or to difference mnemonic judgements, as in the papers the reviewer referred to, in our task, such potential confidence levels, at least as reflected in accuracy or RT, did not influence the connectivity data. We added this following comment after reporting the control analyses for accuracy and RT: “Thus, while some aspects of medial-temporal lobe as well as parietal lobe activity definitely correspond to levels of confidence in mnemonic judgements (Brown, Rissman, Chow, Uncapher, & Wagner, 2018; Hutchinson et al., 2014), in our task such potential confidence levels, as reflected in accuracy and RT, did not influence functional connectivity.”

3) The pattern similarity analyses are a very nice addition and important to the arguments being made. They did leave me wondering about the univariate activity data. To what extent does trial-trial variance change across the different change bins in the data (relevant to the connectivity analyses) and to what extent does CA1 activity change from 0-4? For example, if CA1 activity decreases with the amount of mismatch in the task this could color the attribution of the pattern similarity data prediction errors (e.g., univariate signal is known to influence similarity amplitudes).

Regarding univariate activity data, we now report these data in the manuscript (p. 10) and in the Supplementary Information. Briefly, a linear trend analysis on the average univariate activation in CA1 in each level of changes revealed a significant linear increase ($t_{(18)} = 2.43, p < .05$). No difference was found in CA3 or

ERC univariate activation based on level of changes. We also ran additional analyses controlling for univariate activation in all of our results and the results hold. We thank the reviewer for this comment, as these additional analyses strengthened our manuscript. Below, we detail the analysis for each finding, that were also mentioned previously in the response to the other reviewers.

Regarding control for univariate activation for our main connectivity results, we note that level of changes did not modulate univariate activation in CA3 or in entorhinal cortex. Thus, univariate activity in these regions cannot account for our connectivity findings. And, since CA1 connectivity with these regions was modulated in opposite direction, CA1 activation is also unlikely to account for our connectivity findings. Nevertheless, to respond more directly to the concern about univariate activation and to establish that the connectivity findings are independent of univariate activation findings, we have also now run an additional control analysis: for each pair of regions (CA1-EC/CA1-CA3) we adopted a mixed-effects linear model with connectivity in each level of changes as the explained variable, and as explaining variables we included the average univariate activation per participant in each of the regions in the pair (i.e., CA1 and EC or CA1 and CA3), as well as the interaction of univariate activation. In addition to the univariate activation, as our main contrast of interest we included in the model as an explaining variable either the linear trend contrast (0-4 level of changes were coded as -2,1,0,1,2) or the match < mismatch contrast (0-change was coded as -4, whereas 1-4 level of changes coded as 1). As a reminder, we found that for CA1-EC connectivity, the match < mismatch contrast better accounted for the pattern of connectivity changes thus we included the match < mismatch contrast as our contrast of interest. For CA1-CA3, a linear contrast better accounted for the connectivity changes, thus we included the linear trend contrast as our contrast of interest. Including univariate activation in our models together with these contrasts (i.e., match < mismatch/linear trend) allowed us to test whether the differences in connectivity across levels of changes are significant also when statistically accounting for univariate activation. In these models, the results remain consistent. Specifically, we compared these full models including both our contrast of interest and univariate activity to models that only included univariate activity (additionally, an intercept per participant was included in all models). We found that for both pairs of regions, the models with the contrast of interest significantly explained more variance compared to the models including only univariate activity (CA1-EC: a model including the match < mismatch contrast: $\chi^2 = 14.32$, $p < .001$, AIC or BIC reductions > 10; CA1-CA3: a model including the linear contrast: $\chi^2 = 11.26$, $p < .001$, AIC or BIC reductions > 6). These results show that for both CA1-EC and CA1-CA3

connectivity, the levels of changes significantly explain variance when controlling for univariate activation. Thus, they suggest that our main connectivity findings are unlikely to be explained by univariate activity in any of the three regions involved. We now report this control analysis in the Results (pp. 10-11).

For the correlation of CA1 reinstatement with CA1-entorhinal connectivity, we now additionally control for univariate activation, both during the cue and during the image. Specifically, we conducted a multiple linear regression that included the match < mismatch connectivity contrast as the explained variable, and as explaining variables we included the CA1 reinstatement measure, univariate activation in CA1 during the cue, and during the match images, as well as the match < mismatch univariate contrast in CA1 and in EC (to account for univariate activation for connectivity). Even when including all of these other factors, the reinstatement variable was significant ($t_{(13)} = 1.93, p < .05$). Thus, the correlation is unlikely explained by univariate activation differences between participants. This analysis is reported in the Supplementary Information.

We also conducted two control analyses for the prediction error analysis, which is now reported in the Supplementary Information: First, in line with reviewer #1's suggestion, for each cue, we subtracted from the correlation of the cue and its corresponding image, the average correlation of that same cue to all other images that are in the same type of trial (i.e., same task, and levels of changes), but correspond to other images. Any activation level differences between levels of changes should appear in the other images of the same level of changes, and thus cancel out by the subtraction. Importantly, like in our main analysis, the match > mismatch contrast was significant ($t_{(18)} = 2.60, p = .018$). Second, to further control for univariate activation, we conducted a multiple regression model in which the prediction error match > mismatch contrast per participant was the explained variable, and as explaining variables we took the same match > mismatch contrasts, but computed for the cue activation and the image activation in the left CA1. Note that here, the intercept reflects the effect of interest, namely, whether the match > mismatch difference is significantly different from 0, when controlling for univariate activation. Indeed, the intercept was highly significant ($t_{(16)} = 3.46, p < .004$), suggesting that CA1 multivariate prediction error signal cannot be explained by univariate activation.

Regarding the reviewer's comment about potential differences between levels of changes in trial-by-trial variance, we note that our beta-series correlation approach controls for this variance. Specifically, in this approach, we compute the correlation between trial-estimated activation across pairs of regions. The

calculation of Pearson's correlation coefficient includes the variance in the trial-by-trial activation in both regions of the pair in the denominator. Since we computed the correlation separately for each level of changes, this measure controls for differences in variance between level of changes.

Reviewers' comments:

Reviewer #1 (Remarks to the Author):

In my initial review, I was excited about the high-level ideas in the manuscript, but had several concerns about the results/analyses. The authors have done a commendable job of addressing these concerns. Several of the new control analyses help rule out important alternative accounts of the data (that would have undermined the claims). For example, controlling for univariate activation in the connectivity analyses helps establish that the connectivity measures are not a simple result of univariate responses. Similarly, they have better established the item-specificity of the prediction results, which is key. In general, the new analyses are largely reassuring. Some of the effect sizes (and the sample size) remain modest, but this concern is somewhat tempered by the fact that there were strong priors for most of these analyses.

Reviewer #2 (Remarks to the Author):

I appreciate the authors revisions on their manuscript involving CA1-CA3 / ERC-CA1 interactions in humans during encoding and retrieval. The authors have addressed the issues about univariate analyses and slope differences / multivariate correlations for CA1-ERC vs. CA3-CA1, although the latter were still not as convincing as might be to support the model in my view because they involve one-sided t-tests. I also continue to have some concerns with the use of an MPRAGE to segment subfields.

As I noted previously:

2) The authors reported using MPRAGE anatomical sequences (line 453) and then co-registering a high-resolution EPI to this sequence. However, there are reasons one might expect the MPRAGE to give insufficient signal to noise and contrast to identify the subfields. For one, T2 TSE are considered the gold standard for identifying subfields and an MPRAGE is unlikely to have the differences in contrast/signal needed to discriminate CA1 from CA3/DG on most slices (L. E. Wisse et al., 2016). The concern is that at least some of the weak/non-significant effects could be driven by poor subfield identification.

To me, the approach of using MPRAGEs is inconsistent with the standards of the field in terms of subfield segmentation (Laura EM Wisse, Biessels, & Geerlings, 2014), although I do understand that other groups who have done subfield segmentation have employed some version of this approach. The searchlight approach was certainly helpful here but overall do not overcome some of the issues with insufficient subfield resolution. I also do not find the authors argument that the results could not occur by chance convincing. This is emphasized by the use of one-tailed tests, which both rev #1 and myself had recommended against. The authors argue that with a strong a priori hypothesis, such one-sided tests (and correlations) are reasonable. To me, the authors can't have it both ways. They are arguing that support for the Hasselmo model is ground-breaking and novel yet at the same time, that the model is well established enough to support one sided t-tests/correlation approaches. My view is that there are alternative models which are not really considered. Additionally, I think correlations with small sample sizes and single tailed tests should be avoided. Finally, as I noted in my past review, I am not sure that the results can really address the Hasselmo model. This is accentuated by the fact that CA3-CA1 connectivity did not correlate with CA1 multivariate patterns. As I noted before, pattern completion and separation are almost certainly happening during both encoding and retrieval. This is likely to happen very rapidly and it seems unclear whether BOLD can completely capture these given

the relatively poor temporal resolution.

Thus, while I think the results stand on firmer footing than before, particularly by dealing with univariate issues and correcting some of the statistical issues, I still find elements incomplete and insufficiently convincing for a high impact journal like Nature Communications.

References

Wisse, L. E., Biessels, G. J., & Geerlings, M. I. (2014). A critical appraisal of the hippocampal subfield segmentation package in FreeSurfer. *Frontiers in aging neuroscience*, 6, 261.

Wisse, L. E., Daugherty, A. M., Olsen, R. K., Berron, D., Carr, V. A., Stark, C. E., et al. (2016). A harmonized segmentation protocol for hippocampal and parahippocampal LID - 10.1002/hipo.22671 [doi]. *Hippocampus*.

Reviewer #3 (Remarks to the Author):

I would like to thank the authors for their thorough consideration of my comments. The data presented are both very interesting and allay my concerns.

Reviewer #2 (Remarks to the Author):

I appreciate the authors revisions on their manuscript involving CA1-CA3 / ERC-CA1 interactions in humans during encoding and retrieval. The authors have addressed the issues about univariate analyses and slope differences / multivariate correlations for CA1-ERC vs. CA3-CA1, although the latter were still not as convincing as might be to support the model in my view because they involve one-sided t-tests. I also continue to have some concerns with the use of an MPRAGE to segment subfields.

We thank the reviewer for the positive evaluation of our manuscript and additional analyses. We address the concerns below.

As I noted previously:

2) The authors reported using MPRAGE anatomical sequences (line 453) and then co-registering a high-resolution EPI to this sequence. However, there are reasons one might expect the MPRAGE to give insufficient signal to noise and contrast to identify the subfields. For one, T2 TSE are considered the gold standard for identifying subfields and an MPRAGE is unlikely to have the differences in contrast/signal needed to discriminate CA1 from CA3/DG on most slices (L. E. Wisse et al., 2016). The concern is that at least some of the weak/non-significant effects could be driven by poor subfield identification.

To me, the approach of using MPRAGEs is inconsistent with the standards of the field in terms of subfield segmentation (Laura EM Wisse, Biessels, & Geerlings, 2014), although I do understand that other groups who have done subfield segmentation have employed some version of this approach.

We acknowledge the reviewer's concern that the current gold standard involves segmentation based on MPRAGE and T2 image. However, as we mentioned in our previous response, unfortunately, T2 images were not collected in this study.

Nevertheless, we believe that this is not a concern in the current study for four reasons: (1) The concern about T2 images pertains mainly to the use of automated segmentation protocols while, in the current study, we employed manual segmentation, which one could argue is the gold standard and is, actually, in many ways, superior than any automatic segmentation methods although more labor intensive. The reason it is not more widely used now is because it is labor intensive (2) It is important to note that the T2 image is especially beneficial when trying to distinguish between subfields CA3 and DG, which we do not do in this study. (3) The reviewer is mainly concerned that a couple of our findings which has a relatively weaker p value may be the result of sub-optimal segmentation. The findings in question is the room-reinstatement effect in the left CA1 and its correlation with CA1-

entorhinal connectivity, and the multivariate prediction error analysis in the left CA1. However, in the previous round of revisions, we modified our analyses (following this reviewer's advice), and our results are now statistically stronger. As detailed above, in response to the remaining concerns of the reviewer, we have now removed the correlation of CA1 reinstatement with CA1-entorhinal connectivity from the main text (4) Any bleeding of signal from other ROIs impairs the ability to obtain our results, which are specific to ROIs. Thus, our results were obtained in the face of such potential contamination. We elaborate on each point in turn.

Regarding T2 images, the concerns raised in the papers the reviewer references are unlikely to apply to our methodology. Wisse et al. (2014) addresses problems in automated segmentation in FreeSurfer software. The concerns in that paper were mainly with assumptions that FreeSurfer is making: "*FreeSurfer segmentation, which is based on the subfield distribution in one coronal section in the body of the hippocampus (Van Leemput et al., 2008, 2009) and then used to segment subfields along the complete long axis of the hippocampus. That lead to displacements of hippocampal subfields boundaries*" (Wisse, Biessels, & Geerlings, 2014). Clearly, these issues do not apply to our methodology, since we used manual segmentation based on anatomical markers. As is mentioned in the paper by Wisse et al. (2017) that the reviewer references, manual segmentation is considered the gold standard. Importantly, the well-established procedure (Kirwan, Jones, Miller, & Stark, 2007) we employed relies on matching 8 coronal slices of the hippocampus to an anatomical image (Duvernoy, 1998) and following anatomical markers that are clearly observable on a T1 scan. Thus, the problems mentioned by Wisse et al. (2014) regarding FreeSurfer's automated segmentation do not apply to our study. Further, according to Wisse et al. (2014, 2016), the T2 image is needed especially to distinguish the cornu ammonis (CA) from dentate gyrus (DG), and not to differentiate CA1 from CA3/DG. Indeed, in functional MRI studies as our current study, the absence of T2 image or the lower final resolution of the EPI images typically leads to combining together CA3 and DG into one ROI. This is seen in current studies published in top-tier journals (e.g., Dimsdale-Zucker, Ritchey, Ekstrom, Yonelinas, & Ranganath, 2018, *Nature Communications*; Kyle, Stokes, Lieberman, Hassan, & Ekstrom, 2015, *eLife*; Bakker, Kirwan, Miller, & Stark, 2008, *Science*). As stated in our Methods section (p. 23), we did not argue that we can distinguish CA3 from DG, thus our CA3 ROI indeed probably includes DG (and CA2). This is also clearly laid out in the title of Figure 2 (p. 12).

Nevertheless, in response to the reviewers' comment, we now additionally mention the ROI concern in the Discussion: "*One limitation of the current study is that our CA3 ROI included the CA2 subregion and the dentate gyrus (DG). While this limitation is necessary given the resolution of our data and is shared by many human studies of hippocampal subfields (Dimsdale-Zucker, Ritchey, Ekstrom, Yonelinas, & Ranganath, 2018; Kyle, Stokes, Lieberman, Hassan, & Ekstrom, 2015; Tomparry, Duncan, & Davachi, 2016), recent advances*

in functional imaging such as 7-Tesla MRI scanners, might enable researchers to distinguish between CA3 and DG (Berron et al., 2016). Future work will, thus, be able to further specify the reported connectivity findings” (p. 19).

In addition, we note that the main concern of the reviewer in the previous review, and as reiterated here, was that some of the secondary results we presented as supporting the main connectivity findings might be due to sub-optimal segmentation. Importantly, there was never a concern regarding the main result of the study, namely, the interaction between levels of changes and functional connectivity in CA1-CA3 vs. CA1-ERC. Clearly, this result is sound, using an F-test ($p < .001$, $\eta_p^2 = 0.25$). Regarding the supporting multivariate analyses, namely, the specific-room reinstatement as well as the prediction error multivariate effects in CA1 (see below for more details), the procedure we used following this reviewer’s advice was successful in increasing SNR, and indeed resulted in statistically strong results (see below). We believe these new results reported in the previous response letter should alleviate the reviewer’s concern about ‘weak effects’. As noted above, the correlation between CA1 reinstatement and CA1-entorhinal connectivity was removed from the main text.

Third, and critically, our results show that in the left hemisphere, CA1-ERC connectivity was modulated by changes in the room images in the opposite direction than CA1-CA3 connectivity. Thus, we are certain that the reviewer would agree with us, that any contamination of the regions, if occurred, should obscure our ability to discover such a strong interaction ($p < .001$, $\eta_p^2 = 0.25$). Had we not found an interaction, and, for example, had both regions showed changes in connectivity in the same direction, this could have been attributed to insufficient segmentation. Nevertheless, we did observe a modulation in opposite directions across regions.

Therefore, while we do not wish to argue we have absolutely perfect segmentation, we believe that potential issues with not having a T2 image are unlikely to apply to our study, and that combining together CA3 with DG in the same ROI cannot explain our findings. As we have shown above, our segmentation procedure is well-suited for our investigation, and is clearly sufficient to discover an interaction between regions, as well as multivariate effects. In response to the reviewer’s concerns, we have now removed one result from the main text and modified the Discussion. If the reviewer is still concerned about the segmentation procedure, we ask the reviewer to please explain how sub-optimal segmentation could have yielded our connectivity modulation in opposite directions, or the other reported results, so that we could further address any concern.

Reviewer comment:

The searchlight approach was certainly helpful here but overall do not overcome some of the issues with insufficient subfield resolution.

We were confused about this comment, as we did not perform a searchlight analysis. In our previous response, we explained why a searchlight is not advised for hippocampal subfields. Specifically, a searchlight may not respect the boundaries of oddly shaped subregions of the hippocampus. Further, and more critically, searchlight would require registration of the ROIs to a template for group analysis, which not only distorts the data (and adds the concern that subfields might bleed to one another), but also requires some smoothing, that may occlude representational similarity analysis (RSA) effects (Kriegeskorte, Mur, & Bandettini, 2008).

Thus, we employed the approach of removing non-responsive voxels. Specifically, in our RSA, we removed from the analyses voxels that demonstrated low univariate activation on average during our task (i.e., voxels were removed based on an independent analysis). Excluding non-responding voxels has been previously used in multivariate studies (Chanales, Oza, Favila, & Kuhl, 2017; Favila, Chanales, & Kuhl, 2016), and was also endorsed by the two other expert reviewers. Importantly, this approach was useful in increasing our SNR, as is now reported in the Results. If the reviewer meant in this comment that our approach was useful, we thank the reviewer for this positive evaluation. However, the reviewer also comments that a searchlight might also not overcome some resolution issues. We ask the reviewer to please clarify the comment about the resolution. Our anatomical scan was of 1^3 mm resolution, which is prevalent and acceptable in functional and anatomical investigations of hippocampal subfields (Dimsdale-Zucker et al., 2018; Iglesias et al., 2015; Kyle et al., 2015; Wisse et al., 2014, 2017). Our in-plane EPI resolution is 1.5^2 mm, which is also a standard for hippocampal subfields studies (Dimsdale-Zucker et al., 2018; Kyle et al., 2015; Tompariy et al., 2016; Schlichting, Zeithamova, & Preston, 2014; note that a 2^2 mm was used in the last study). Thus, we feel positive that our resolution is well adequate to our investigation.

I also do not find the authors argument that the results could not occur by chance convincing. This is emphasized by the use of one-tailed tests, which both rev #1 and myself had recommended against. The authors argue that with a strong a priori hypothesis, such one-sided tests (and correlations) are reasonable. To me, the authors can't have it both ways. They are arguing that support for the Hasselmo model is ground-breaking and novel yet at the same time, that the model is well established enough to support one sided t-tests/correlation approaches.

We wish to clarify that, putting aside one result that we now removed from the main text as detailed above, all of the results reported in the manuscript are statistically strong using F-tests or two-tailed t-tests. Specifically, regarding our main functional connectivity result, the interaction between Changes (0-4) and ROI (entorhinal, CA3) was tested using a repeated measures ANOVA, and was highly significant ($F_{(4,72)} = 6.04$, $p < .001$, $\eta_p^2 = 0.25$). Follow-up analyses were also conducted using ANOVAs. Specifically, two repeated-measures ANOVAs were conducted, one for each pair of regions, with the factor of Changes (0-4). These ANOVAs

confirmed that in both regions pairs (CA1-entorhinal or CA1-CA3), Changes significantly modulated connectivity (CA1-entorhinal: $F_{(4,72)} = 4.49$, $p < .003$, $\eta_p^2 = 0.20$; CA1-CA3: $F_{(4,72)} = 3.58$, $p < .02$, $\eta_p^2 = 0.17$).

The additional results are supportive of our interpretation that mnemonic prediction errors modulate hippocampal connectivity, and are also statistically sound. Our first supportive result show room-specific reinstatement in the left CA1. Specifically, we correlated, for each cue, the multivoxel activity pattern during the cue with the activity pattern of the match image (the 0-changes image) of the corresponding room, and averaged across cues. We then compared that same-room similarity to the similarity of the cue to match images corresponding to other rooms, to obtain a measure of room-specific reinstatement (as was done in the original manuscript). We found that the similarity of the cue to the corresponding room image was significantly higher than the similarity to other room images. Although we have originally reported this result using a one-tailed t-test, as we mentioned in our previous response, and as is clearly evident by looking at the result of the statistical test, this result is significant also when employing a two-tailed test. Acknowledging the reviewer's concern, we now report this result using a two-tailed t-test (we also include now Cohen's d , which we unfortunately omitted in our revision): *"We found that the correlation with the corresponding room was higher than with the other rooms suggesting that specific-room reinstatement took place in the left CA1 (match: $M = .01$, $SD = 0.01$; other: $M = .0045$, $SD = .005$; $t(18) = 2.5$, $p = .02$; Cohen's $d = .57$)"* (p. 13).

The second supportive result is the multivariate CA1 prediction error analysis. In this analysis, we calculated the similarity between the activity patterns in the left CA1 during the cue and during the image, separately for each level of changes. As is reported in the Results, we found that pattern similarity in CA1 decreased when we introduced changes in the images (the match > mismatch contrast was highly significant: $t_{(18)} = 3.14$, $p < .006$, two-tailed, Cohen's $d = .72$; p. 14).

To sum up, all of the results are currently reported using F-tests or two-tailed t-tests, and are highly significant. As the effect-size estimates above show, we report medium to large effects. If the reviewer still does not find the strength of these results convincing, please say why, and we will address any additional concerns.

Although in the current version all of the results in the main text are reported using an F-test or two-tailed t-test, we would like to further address the reviewers' comment about our investigation of the theoretical model. In our view, in empirical sciences, a model, as established as may be, is still only a theoretical suggestion awaiting empirical evidence. Naturally, we strongly support rigorous theoretical models and this current investigation is motivated by such models. It is important, however, especially for prominent models, to identify parts that have yet not been supported by empirical evidence. In humans, many aspects of the Hasselmo model and other models have yet to be empirically tested (but see,

e.g., Duncan, Sadanand, & Davachi, 2012; Duncan, Tomparry, & Davachi, 2014; Griffiths et al., 2019). Specifically, the conjunction between manipulating mnemonic prediction error or novelty, and testing functional connectivity between hippocampal subfields is novel with respect to the model – as there is no empirical evidence for it. We further note that our investigation is not specific to the Hasselmo model, but pertains to other models and theoretical frameworks implicating the hippocampus in novelty and prediction error (e.g., Colgin, 2016; Henson & Gagnepain, 2010; Kafkas & Montaldi, 2018; Kumaran & Maguire, 2007; Lisman & Grace, 2005; McClelland, McNaughton, & O'Reilly, 1995; Meeter, Murre, & Talamini, 2004). Thus, our findings are novel and relevant to a broad community of researchers. The results reported in the paper are also novel and significant with respect to the long-lasting interest in prediction errors across domains of neuroscience and psychology (e.g., Bar, 2009; Friston, 2018; Heeger, 2017; Rescorla & Wagner, 1972; Schultz, Dayan, & Montague, 1997). Further, showing that connectivity between hippocampal subfields is modulated by task demands more generally is novel, as there is only scarce evidence for that (e.g., Duncan et al., 2014; Tomparry, Duncan, & Davachi, 2015). Thus, our findings are groundbreaking, and statistically strong.

Reviewer comment:

My view is that there are alternative models which are not really considered.

We note that we have considered alternative models and hypotheses. Specifically, any statistical investigation is testing against an alternative model. In the current study, the ANOVA's we conducted (as detailed above, and in the Methods, p. 27 and Results, pp. 9-10), successfully tested our hypothesis against the hypothesis that mnemonic prediction errors do not influence hippocampal connectivity. Another model that is implicitly tested is that mnemonic prediction errors uniformly increase connectivity in the hippocampus, regardless of the subfield specificity. Such a model would not predict an interaction, as we have found. We did not incorporate this possibility in the Introduction in the interest of streamlining the Introduction, but we do address it in the Discussion: *“Thus, mnemonic prediction errors do not simply lead to an overall general increase (or decrease) in functional connectivity of the CA1 region, but rather they selectively and differentially modulate processing along distinct hippocampal pathways.”* (p. 16). Additionally, and although this was not the main aim of the study, we further tested models characterizing the increase or decrease in connectivity as linear (namely, a linear change across levels of changes, 0-4), or as corresponding to a match vs. mismatch signal (i.e., all levels of changes 1-4 are different from no changes; see Methods, p. 27 and Results, pp. 10-11). Moreover, following Reviewer #3's comments, we have also tested whether a model based on behavioral responses (specifically, accuracy and reaction times) might account for the results better than levels of changes in the images. We did not find evidence supporting these models, and we found that the level of changes significantly

explained variance even when statistically controlling for behavioral factors (see Supplementary Information). Thus, we have considered a number of alternative models. If the reviewer still believes that there are alternative models that were not considered and should be considered in the current paper, we encourage the reviewer to suggest these alternative models, and we are happy to test them. If these models would account for the data better than the current models, we would be the first to thank the reviewer for useful suggestions, and we would certainly modify the manuscript accordingly.

Reviewer Comment:

Additionally, I think correlations with small sample sizes and single tailed tests should be avoided.

Thank you. As mentioned above, we have now moved the correlation to the Supplementary Information. We have additionally removed mentions of this correlation from the main text in several places, in favor of emphasizing the main effects observed in our RSA analyses, namely, the prediction strength and prediction error findings:

1. In the abstract, we removed a sentence referring to the correlation, and added the following sentence: *“Further, in the left CA1, the similarity between activity patterns corresponding to retrieval of the learned room during the cue, and activity patterns during the image, was lower when the image included changes, consistent with a prediction error signal in CA1.”* (p. 2).
2. Towards the end of the Introduction, the sentence addressing the correlation was removed, and we added the following: *“Moreover, CA1 multivoxel activity patterns during the cue showed evidence of room-specific retrieval, potentially reflecting prediction of the learned room. Further, introducing changes in the room image led to lower similarity between CA1 activity patterns during memory retrieval at the cue and during viewing the room image, consistent with a mnemonic prediction error signal in CA1.”* (p. 6).
3. The Results section describing the room-specific reinstatement is now titled: *“CA1 multivoxel activity patterns during the cue show evidence of room-specific predictions”* (p. 13) and is opened with the following: *“In the previous analysis, mnemonic prediction errors were operationalized as changes in the probe room image, relative to a retrieved memory of that room. Here, we sought to support the notion that our task involved mnemonic prediction errors, by providing evidence for room-specific predictions in CA1 during the cue, that is, when participants were asked to retrieve the room.”* (p. 13). The correlation with connectivity is only mentioned briefly, and the reader is referred to the supplementary: *“Interestingly, we have also found a moderate correlation across participants between prediction-strength in CA1 and the increase in CA1-entorhinal connectivity in response to errors (see Supplementary Information, and supplementary*

figure S2), lending further support for the notion that functional connectivity increases are related to predictions and their violations. “ (p. 13).

4. The Results section describing the mnemonic-prediction error RSA previously opened with a note of the correlation, but now starts with these sentences: *“The previous result suggests that CA1 activity patterns during the cue capture participants’ predictions. However, it does not directly examine participants’ prediction errors. “ (p. 13).*
5. The third paragraph of the Discussion, addressing the supporting findings, now includes the following sentences: *“To support the notion that connectivity changes were related to participants’ internal memory predictions, we quantified prediction strength and mnemonic prediction-error by examining the multi-voxel activity patterns in CA1. We found higher similarity between activity patterns corresponding a retrieved memory of a room and viewing of that same room compared to other rooms, indicating room-specific memory reinstatement, or prediction, in CA1. Further, as a supplemental finding, we see evidence that participants with better cued memory reinstatement showed a greater increase in CA1-entorhinal connectivity in response to subsequent violations of the remembered rooms. ... We have additionally found that in CA1, activity patterns during cued memory reinstatement were more similar to activity patterns during viewing the same image, compared to viewing an altered version of image. This result is consistent with the notion that CA1 activity patterns are sensitive to mnemonic prediction errors.“ (pp. 16-17).*
6. Later in the Discussion, after discussing the potential involvement of VTA projections to CA1 and entorhinal cortex in producing changes in connectivity, we now comment: *“...Consistent with that notion, we show preliminary results suggesting that connectivity in CA1-entorhinal cortex was correlated with the strength of the memory predictions measured in area CA1”*. The words *“preliminary results suggesting”* were added to further qualify this finding (p. 18).

Reviewer Comment:

Finally, as I noted in my past review, I am not sure that the results can really address the Hasselmo model. This is accentuated by the fact that CA3-CA1 connectivity did not correlate with CA1 multivariate patterns. As I noted before, pattern completion and separation are almost certainly happening during both encoding and retrieval. This is likely to happen very rapidly and it seems unclear whether BOLD can completely capture these given the relatively poor temporal resolution.

First, as noted above, while we acknowledge the importance and impact of the Hasselmo model and are clearly motivated by it, we do not see our results as pertaining to this model alone but rather to a broad cohort of models and theoretical work addressing hippocampal processing (e.g., Colgin, 2016; Henson & Gagnepain, 2010; Kafkas & Montaldi, 2018; Kumaran & Maguire, 2007; Lisman & Grace, 2005; McClelland et al., 1995; Meeter et al., 2004).

Second, and importantly, the Hasselmo model as well as some of the models mentioned above specifically address communication between hippocampal subfields, and predict that novelty, or prediction error, should increase connectivity between CA1 and entorhinal cortex, and decrease connectivity between CA1 and CA3 (Hasselmo, Bodelón, & Wyble, 2002; Hasselmo & Stern, 2014). Thus, the critical test for these models is the main result of the manuscript, namely, that changes in perceptual input relative to a retrieved memory modulate connectivity in the hippocampus. The connectivity pattern that we observed, namely, a reduction in CA1-CA3 functional connectivity and an increase in CA1-entorhinal cortex connectivity, depict the pattern of connectivity that the Hasselmo model and other models predict. Thus, these results clearly provide novel evidence for these models, and greatly advance our understanding of hippocampal functioning.

We also wish to emphasize that fMRI and the measured BOLD signal are adequate to test functional connectivity (Rissman, Gazzaley, & D'Esposito, 2004), and especially between hippocampal subfields that require high spatial resolution (Duncan et al., 2014; Tomparry et al., 2015). The additional supportive result of the correlation between CA1-entorhinal connectivity and CA1 multivariate patterns therefore was not the primary test of the model and was meant to support our argument (we also now have removed this data point to the Supplementary Information).

Naturally, as any study, we did not aim to test all components of the model, and specifically, we agree that we cannot address pattern completion or separation in the hippocampus. We do not believe that one study could, or should, necessarily address all aspects of any model. Therefore, in our view, whether we tested all components of a certain model or not should not be a criterion for publication. Our investigation was directed towards testing predictions of the Hasselmo as well as other hippocampal models. Nevertheless, the point of the reviewer is well taken. To clarify our interpretation even more and facilitate future research, we now explicitly discuss this point in the Discussion: *“Another intriguing question for further research is whether and how functional connectivity between hippocampal subregions is related to pattern completion versus pattern separation mechanisms (Knierim & Neunuebel, 2016; Marr, 1971; O’Reilly & McClelland, 1994; Treves & Rolls, 1994; Yassa & Stark, 2011). Pattern completion, namely, the reinstatement of a previously encoded activity pattern, has been suggested to underlie memory retrieval (Grande et al., 2019; Knierim & Neunuebel, 2016; Neunuebel & Knierim, 2014; Staresina et al., 2016). Pattern separation, or the allocation of a distinct activity pattern to similar experiences, is suggested to underlie encoding of new memories (Bakker et al., 2008; Berron et al. 2016; Favila et al., 2015; Leutgeb et al. 2007). Likewise, previous empirical work has provided evidence that communication between CA1 and CA3, and between CA1 and entorhinal cortex may support retrieval versus encoding, correspondingly (e.g., Duncan et al., 2014; Fernández-Ruiz et al., 2017; Montgomery & Buzsaki, 2007; Schomburg et al., 2014; Tort et al., 2009). The current study, however, did not aim to directly test whether pattern completion or*

separation took place as a result of mnemonic prediction errors, but to examine whether memory violations are related to changes in intrahippocampal connectivity. Thus, the specific computations mediated by communication between hippocampal subregions or by the subregions themselves in the case of mnemonic prediction errors, are exciting questions for future research to explore.” (pp. 18-19).

References

- Bakker, A., Kirwan, C. B., Miller, M., & Stark, C. E. L. (2008). Pattern Separation in the Human Hippocampal CA3 and Dentate Gyrus, *319*, 1640–1643. <https://doi.org/10.1126/science.1152882>
- Bar, M. (2009). The proactive brain: memory for predictions. *Philosophical Transactions of the Royal Society of London. Series B, Biological Sciences*, *364*(1521), 1235–1243. <https://doi.org/10.1098/rstb.2008.0310>
- Berron, D., Schutze, H., Maass, A., Cardenas-Blanco, A., Kuijf, H. J., Kumaran, D., & Duezel, E. (2016). Strong Evidence for Pattern Separation in Human Dentate Gyrus. *Journal of Neuroscience*, *36*(29), 7569–7579. <https://doi.org/10.1523/JNEUROSCI.0518-16.2016>
- Chanales, A. J. H., Oza, A., Favila, S. E., & Kuhl, B. A. (2017). Overlap among Spatial Memories Triggers Repulsion of Hippocampal Representations Article Overlap among Spatial Memories Triggers Repulsion of Hippocampal Representations. *Current Biology*, *27*(15), 2307–2317. <https://doi.org/10.1016/j.cub.2017.06.057>
- Colgin, L. L. (2016). Rhythms of the hippocampal network. *Nature Reviews Neuroscience*, *17*(4), 239–249. <https://doi.org/10.1038/nrn.2016.21>
- Dimsdale-Zucker, H. R., Ritchey, M., Ekstrom, A. D., Yonelinas, A. P., & Ranganath, C. (2018). CA1 and CA3 differentially support spontaneous retrieval of episodic contexts within human hippocampal subfields. *Nature Communications*, *9*(1). <https://doi.org/10.1038/s41467-017-02752-1>
- Duncan, K., Sadanand, A., & Davachi, L. (2012). Memory’s Penumbra: Episodic memory decisions induce lingering mnemonic biases. *Science*, *337*(6093), 485–487. <https://doi.org/10.1126/science.1221936>
- Duncan, K., Tompary, A., & Davachi, L. (2014). Associative Encoding and Retrieval Are Predicted by Functional Connectivity in Distinct Hippocampal Area CA1 Pathways. *Journal of Neuroscience*, *34*(34), 11188–11198. <https://doi.org/10.1523/JNEUROSCI.0521-14.2014>
- Duvernoy, H. (1998). *The Human Hippocampus*. New York: Springer-Verlag.
- Favila, S. E., Chanales, A. J. H., & Kuhl, B. A. (2016). Experience-dependent hippocampal pattern differentiation prevents interference during subsequent learning. *Nature Communications*, *6*, 1–10. <https://doi.org/10.1038/ncomms11066>
- Friston, K. J. (2018). Does predictive coding have a future? *Nature Neuroscience*, *21*(8), 1019–1021. <https://doi.org/10.1038/s41593-018-0200-7>
- Griffiths, B. J., Parish, G., Roux, F., Michelmann, S., van der Plas, M., Kolibius, L. D., ... Hanslmayr, S. (2019). Directional coupling of slow and fast hippocampal gamma with neocortical alpha/beta oscillations in human episodic memory. *Proceedings of the National Academy of Sciences of the United States of America*, 1–9.

<https://doi.org/10.1073/pnas.1914180116>

- Hasselmo, M. E., Bodelón, C., & Wyble, B. P. (2002). A proposed function for hippocampal theta rhythm: Separate phases of encoding and retrieval enhance reversal of prior learning. *Neural Computation*, *14*(4), 793–817. <https://doi.org/10.1162/089976602317318965>
- Hasselmo, M. E., & Stern, C. E. (2014). Theta rhythm and the encoding and retrieval of space and time. *NeuroImage*, *85*, 656–666. <https://doi.org/10.1016/j.neuroimage.2013.06.022>
- Heeger, D. J. (2017). Theory of cortical function. *Proceedings of the National Academy of Sciences*, *114*(8), 1773–1782. <https://doi.org/10.1073/pnas.1619788114>
- Henson, Richard N., & Gagnepain, P. (2010). Predictive, Interactive Multiple Memory Systems. *Hippocampus*, *20*(11), 1315–1326. <https://doi.org/10.1002/hipo.20857>
- Iglesias, J. E., Augustinack, J. C., Nguyen, K., Player, C. M., Player, A., Wright, M., ... Van Leemput, K. (2015). A computational atlas of the hippocampal formation using ex vivo, ultra-high resolution MRI: Application to adaptive segmentation of in vivo MRI. *NeuroImage*, *115*, 117–137. <https://doi.org/10.1016/j.neuroimage.2015.04.042>
- Kafkas, A., & Montaldi, D. (2018). How do memory systems detect and respond to novelty? *Neuroscience Letters*, (January), 0–1. <https://doi.org/10.1016/j.neulet.2018.01.053>
- Kirwan, C. B., Jones, C. K., Miller, M. I., & Stark, C. E. L. (2007). High-resolution fMRI investigation of the medial temporal lobe. *Human Brain Mapping*, *28*(10), 959–966. <https://doi.org/10.1002/hbm.20331>
- Kriegeskorte, N., Mur, M., & Bandettini, P. (2008). Representational similarity analysis - connecting the branches of systems neuroscience. *Frontiers in Systems Neuroscience*, *2*, 4. <https://doi.org/10.3389/neuro.06.004.2008>
- Kumaran, D., & Maguire, E. A. (2007). Which Computational Mechanism Operate in the Hippocampus During Novelty Detection? *Hippocampus*, *17*, 735–748. <https://doi.org/10.1002/hipo>
- Kyle, C. T., Stokes, J. D., Lieberman, J. S., Hassan, A. S., & Ekstrom, A. D. (2015). Successful retrieval of competing spatial environments in humans involves hippocampal pattern separation mechanisms, 1–19. <https://doi.org/10.7554/eLife.10499>
- Lisman, J. E., & Grace, A. A. (2005). The hippocampal-VTA loop: Controlling the entry of information into long-term memory. *Neuron*, *46*(5), 703–713. <https://doi.org/10.1016/j.neuron.2005.05.002>
- McClelland, J. L., McNaughton, B. L., & O'Reilly, R. C. (1995). Why there are complementary learning-systems in the hippocampus and neocortex – insights from the success and failures of connectionist models of learning and memory. *Psychological Review*, *102*(3), 419–457. <https://doi.org/10.1037/0033-295x.102.3.419>
- Meeter, M., Murre, J. M. J., & Talamini, L. M. (2004). Mode shifting between storage and recall based on novelty detection in oscillating hippocampal circuits. *Hippocampus*, *14*(6), 722–741. <https://doi.org/10.1002/hipo.10214>
- Rescorla, R. A., & Wagner, A. R. (1972). A theory of Pavlovian conditioning. *Classical Conditioning II Current Research and Theory*. <https://doi.org/10.1101/gr.110528.110>
- Rissman, J., Gazzaley, A., & D'Esposito, M. (2004). Measuring functional connectivity during distinct stages of a cognitive task. *NeuroImage*, *23*(2), 752–763. <https://doi.org/10.1016/j.neuroimage.2004.06.035>
- Schlichting, M. L., Zeithamova, D., & Preston, A. R. (2014). CA<inf>1</inf> subfield contributions

- to memory integration and inference. *Hippocampus*, 24(10), 1248–1260.
<https://doi.org/10.1002/hipo.22310>
- Schultz, W., Dayan, P., & Montague, P. R. (1997). A neural substrate of prediction and reward. *Science*, 275(June 1994), 1593–1599. <https://doi.org/10.1126/science.275.5306.1593>
- Tompary, A., Duncan, K., & Davachi, L. (2015). Consolidation of Associative and Item Memory Is Related to Post-Encoding Functional Connectivity between the Ventral Tegmental Area and Different Medial Temporal Lobe Subregions during an Unrelated Task. *The Journal of Neuroscience : The Official Journal of the Society for Neuroscience*, 35(19), 7326–7331.
<https://doi.org/10.1523/JNEUROSCI.4816-14.2015>
- Tompary, A., Duncan, K., & Davachi, L. (2016). High-resolution investigation of memory-specific reinstatement in the hippocampus and perirhinal cortex. *Hippocampus*, 26(8), 995–1007.
<https://doi.org/10.1002/hipo.22582>
- Wisse, L. E. M., Biessels, G. J., & Geerlings, M. I. (2014). A critical appraisal of the hippocampal subfield segmentation package in FreeSurfer. *Frontiers in Aging Neuroscience*, 6, 261.
- Wisse, L. E. M., Daugherty, A. M., Olsen, R. K., Berron, D., Carr, V. A., Stark, C. E. L., ... la Joie, R. (2017). A harmonized segmentation protocol for hippocampal and parahippocampal subregions: Why do we need one and what are the key goals? *Hippocampus*, 27(1), 3–11.
<https://doi.org/10.1002/hipo.22671>

***REVIEWERS' COMMENTS:

Reviewer #1 (Remarks to the Author):

This is a supplemental review to comment to the authors' response to another reviewer's concerns:

I've read through R2's remaining concerns and I do think they are valid. I don't disagree with any of the points. In my first review, I noted the small sample size and the use of 1-tailed t-tests. I had not thought about the point raised by R2 that sub-field segmentation was based on an MPRAGE. R2 is certainly correct that this is not up to current standards. Put another way, if someone was designing a study to test hippocampal sub-field predictions, you would certainly include an anatomical scan geared toward this. But this is a re-analysis of a fairly old data set—prior to current norms. This paper has some notable strengths (it is nicely written, has some interesting ideas), but I do think there are several places, as R2 notes (and as I noted in my initial review) where this paper feels like it just isn't up to current standards.

1. For fMRI studies, it is increasingly uncommon, particularly in high impact journals, to have an $n < 20$. Even an $n < 30$ is sometimes described as low. This paper has an n of 19. And between-subjects correlations (as reported here) with small n 's have been the subject of particular scrutiny.
2. The authors defend the use of one-tailed tests and, from a statistics perspective, their argument is fine. But, overwhelmingly, the field uses two-tailed tests even when one-tailed tests can be justified. I think the vast majority of readers view the one-tailed tests as something you pull out if needed. So, again, this is just not the norm.
3. Again, the sub-field segmentation is not wrong, but is just not up to current standards.

Reviewer #3 (Remarks to the Author):

It is my view that the authors have done a satisfactory job addressing the remaining concerns from R2. In particular, the authors have better highlighted the analyses on which their principal conclusions are drawn over those that should be considered "supplemental", have better justified the application of their approaches with T1-weighted MRI images and acknowledged the associated limitations, and have further contextualized their predictions and outcomes with different models of MTL subdivision function.

REVIEWERS' COMMENTS:

Reviewer #1 (Remarks to the Author):

This is a supplemental review to comment to the authors' response to another reviewer's concerns:

I've read through R2's remaining concerns and I do think they are valid. I don't disagree with any of the points. In my first review, I noted the small sample size and the use of 1-tailed t-tests. I had not thought about the point raised by R2 that sub-field segmentation was based on an MPRAGE. R2 is certainly correct that this is not up to current standards. Put another way, if someone was designing a study to test hippocampal sub-field predictions, you would certainly include an anatomical scan geared toward this. But this is a re-analysis of a fairly old data set—prior to current norms. This paper has some notable strengths (it is nicely written, has some interesting ideas), but I do think there are several places, as R2 notes (and as I noted in my initial review) where this paper feels like it just isn't up to current standards.

1. For fMRI studies, it is increasingly uncommon, particularly in high impact journals, to have an $n < 20$. Even an $n < 30$ is sometimes described as low. This paper has an n of 19. And between-subjects correlations (as reported here) with small n 's have been the subject of particular scrutiny.

We appreciate these concerns. Regarding the sample size, for each result, we have now added the number of participants showing the group-level trend. As can be seen, the vast majority of our participants demonstrated the reported effects, strengthening our confidence that our sample is representative, and that our results do not stem from a singular or even a few participants swaying the distribution.

As a reminder, we found an increase in functional connectivity between CA1-entorhinal cortex, and a concomitant decrease in CA1-CA3 connectivity, as the number of changes in the room-images increased. In the results section describing these results, we

added: *“We further examined how many individual participants demonstrated the trends reported at the group level. We computed, for each participant, the match < mismatch contrast in CA1-entorhinal connectivity and the linear contrast in CA1-CA3 connectivity (contrasts are as described above and in the Methods), as these were the contrast revealed to best explain variance in functional connectivity in these regions. Indeed, the overwhelming majority of the participants showed a positive match < mismatch trend in CA1-entorhinal connectivity (16 out of 19 participants, 84% of the participants), and a negative linear trend in CA1-CA3 connectivity (18 out of 19 participants, 95% of the participants).”* (p. 10).

As a supportive result, we examine and report evidence for room-specific reinstatement in area CA1. Namely, we see higher similarity between CA1 multivoxel activity patterns during the retrieval cue and during viewing of the same cued room, compared to other rooms. Along with the results of the statistical tests for this effect, we now report: *“14 out of 19 participants, which are 74% of our participants, demonstrated quantitatively higher same- vs. other-rooms reinstatement”* (p. 13). An additional supportive result is that pattern similarity is lower between CA1 activity patterns during the cue and during the room image presentation of each trial, when the image includes changes, compared to an identical image to the learned room. Here, we added to the report of the statistics: *“14 out of 19 participants, which are 74% of our participants, demonstrated a positive match > mismatch contrast”*. (p. 14).

The across-participant correlation refers to the finding whereby participants who showed greater evidence of our measure of a mnemonic prediction tended to exhibit larger increases in CA1-entorhinal connectivity in response to mismatching rooms. In response to the concerns, we had moved this result to the Supplementary Information and added discussion that this was to be considered preliminary evidence (see below). We also provided a figure for full transparency of each data point so that readers could appreciate that the correlation does not stem from outliers, which could be a concern with across-participant correlations with moderate N. In addition, in the Discussion, we explicitly noted: *“We show preliminary results suggesting that connectivity in CA1-entorhinal cortex was correlated with the strength of the memory predictions measured in area CA1”* (p. 17). We note, and as detailed in the

Supplementary Information, that each data point is a within-participant measure. Specifically, room-specific reinstatement is the difference between same-room vs. other-rooms reinstatement. The match < mismatch contrast score that we used for CA1-enotrinal cortex connectivity is the difference between the 0-changes condition and the average of all levels of changes. Thus, this correlation cannot stem from differences in any baseline levels of our participants. Given the points mentioned here, and since the reviewer acknowledges that our procedure is statistically valid (see point 2), we prefer to provide these data, with the qualification, rather than excluding it from the manuscript. As we mentioned in previous revisions, we are happy to exclude this data point as well.

2. The authors defend the use of one-tailed tests and, from a statistics perspective, their argument is fine. But, overwhelmingly, the field uses two-tailed tests even when one-tailed tests can be justified. I think the vast majority of readers view the one-tailed tests as something you pull out if needed. So, again, this is just not the norm.

We thank the reviewer for acknowledging that our procedure is statistically sound. As detailed above, we only regard this finding as preliminary. Therefore, these data are reported in the Supplementary Information, and we explicitly discuss them as preliminary.

3. Again, the sub-field segmentation is not wrong, but is just not up to current standards.

We now specifically address this concern in the Discussion. In the paragraph addressing the limitation of combining together the CA3 and DG ROI, we added: *“Relatedly, we note that we based our manual segmentation on a T1-weighted image, while some recent proposals recommend using an additional T2-weighted image⁷⁰. The main advantage in using a T2 image is in distinguishing the cornu ammonis (CA) from dentate gyrus (DG)^{70,71}. As mentioned above, we did not aim to distinguish CA3 from DG in the current study. Our manual segmentation was performed based on a well-established procedure relying on markers clearly visible on a T1 image^{72,73}, thus we believe that this concern is unlikely to influence our results. Importantly, we report that functional connectivity between*

hippocampal subfields was modulated in opposite directions: CA1-CA3 connectivity increased, while CA1-entorhinal cortex connectivity increased, as number of changes in the images increased. Any possible blurring of ROIs, if occurred, would therefore impair our ability to observe this dissociation.” (pp. 18-19).

Reviewer #3 (Remarks to the Author):

It is my view that the authors have done a satisfactory job addressing the remaining concerns from R2. In particular, the authors have better highlighted the analyses on which their principal conclusions are drawn over those that should be considered "supplemental", have better justified the application of their approaches with T1-weighted MRI images and acknowledged the associated limitations, and have further contextualized their predictions and outcomes with different models of MTL subdivision function.

We thank the reviewer for the positive evaluation of our work.